# Setting the Record Straight on Transformer Oversmoothing

**Gbètondji J-S Dovonon**                                    *gbetondji.dovonon.22@ucl.ac.uk*
*University College London*

**Michael Bronstein**                                        *michael.bronstein@cs.ox.ac.uk*
*University of Oxford*

**Matt J. Kusner**                                           *matt.kusner@mila.quebec*
*Polytechnique Montréal, Mila — Quebec AI Institute*

**Reviewed on OpenReview:** *https://openreview.net/forum?id=HHI6qWLFF1*

## Abstract

Transformer-based models have recently become wildly successful across a diverse set of domains. At the same time, recent work has shown empirically and theoretically that Transformers are inherently limited. Specifically, they argue that as model depth increases, Transformers oversmooth, i.e., inputs become more and more similar. A natural question is: How can Transformers achieve these successes given this shortcoming? In this work we test these observations empirically and theoretically and uncover a number of surprising findings. We find that there are cases where feature similarity increases but, contrary to prior results, this is not inevitable, even for existing pre-trained models. Theoretically, we show that smoothing behavior depends on the eigenspectrum of the value and projection weights. We verify this empirically and observe that the sign of layer normalization weights can influence this effect. Our analysis reveals a simple way to parameterize the weights of the Transformer update equations to influence smoothing behavior. We hope that our findings give ML researchers and practitioners additional insight into how to develop future Transformer-based models.

## 1 Introduction

In recent years, Transformer models (Vaswani et al., 2017) have achieved astounding success across vastly different domains: e.g., vision (Dosovitskiy et al., 2021; Touvron et al., 2021a), NLP (Touvron et al., 2023; Wei et al., 2023; Kaddour et al., 2023a), chemistry (Schwaller et al., 2019), and many others. However their performance can quickly saturate as model depth increases (Kaplan et al., 2020; Wang et al., 2022). This appears to be caused by fundamental properties of Transformer models. Empirically, researchers first observed that as depth was increased, even to just 12 layers, features became more and more similar to one another (Tang et al., 2021; Zhou et al., 2021a;b; Gong et al., 2021; Yan et al., 2022). Theoretically, these observations were characterized as (a) **Input Convergence**: Transformer features converge to the exact same vector (Park & Kim, 2022; Wang et al., 2022; Bai et al., 2022); (b) **Angle Convergence**: the angle between Transformer features converges to 0 (Tang et al., 2021; Zhou et al., 2021a; Gong et al., 2021; Yan et al., 2022; Shi et al., 2022; Noci et al., 2022; Guo et al., 2023); or (c) **Rank Collapse**: Transformer features collapse to a rank one matrix (Dong et al., 2021; Shi et al., 2022; Noci et al., 2022; Guo et al., 2023; Ali et al., 2023). In practice, this has led to a search for replacements for Transformer layers, including completely new attention blocks (Zhou et al., 2021a;b; Wang et al., 2022; Ali et al., 2023), normalization layers (Guo et al., 2023; Zhai et al., 2023), altered skip connections (Tang et al., 2021; Noci et al., 2022; Shi et al., 2022), convolutional layers (Park & Kim, 2022), fully-connected layers (Liu et al., 2021a; Kocsis et al., 2022; Yu et al., 2022a), and even average pooling layers (Yu et al., 2022b).

But are Transformers destined to oversmooth? In this work we test the above observations theoretically and empirically. Theoretically, we analyze the eigenspectrum of a simplified Transformer layer: fixed attention,

weights, and a residual connection. We show even for this simplified setup that: (a) There are cases where all features converge to the same vector, but this is not inevitable, contrary to prior results; (b) Angle convergence is also possible, but not guaranteed; and (c) while rank collapse is likely, it is also not necessarily happening. Empirically, for existing pre-trained models we find cases where (a) features do not converge to the same vector, (c) feature angles do not converge to 0, and (c) rank does not collapse. In fact, our analysis uncovers a parameterization that allows one, in some cases better than others, to increase smoothing or reduce it. We observe that the sign of the weights of layer normalization plays a role in how much this parameterization influences smoothing behavior.

## 2 Background & Related Work

### 2.1 The Transformer Update

At their core, Transformers are a linear combination of a set of 'heads'. Each head applies its own self-attention function on the input $\mathbf{X} \in \mathbb{R}^{n \times d}$ as follows

$$\mathbf{A} := \mathsf{Softmax}\Big(\frac{1}{\sqrt{k}}\mathbf{X}\mathbf{W}_Q\mathbf{W}_K^\top\mathbf{X}^\top\Big), \tag{1}$$

where the $\mathsf{Softmax}(\cdot)$ function is applied to each row individually. Further, $\mathbf{W}_Q, \mathbf{W}_K \in \mathbb{R}^{d \times k}$ are learned query and key weight matrices. This 'attention map' $\mathbf{A}$ then transforms the input to produce the output of a single head: $\mathbf{A}\mathbf{X}\mathbf{W}_V\mathbf{W}_{\mathsf{proj}}$, where $\mathbf{W}_V, \mathbf{W}_{\mathsf{proj}} \in \mathbb{R}^{d \times d}$ are learned value and projection weights. Most architectures then add a residual connection:

$$\mathbf{X}_\ell = \mathbf{X}_{\ell-1} + \mathbf{A}\mathbf{X}_{\ell-1}\mathbf{W}_V\mathbf{W}_{\mathsf{proj}}. \tag{2}$$

These architectures also consist of layer-specific attention and weights, multiple heads (i.e., multiple $\mathbf{A}, \mathbf{W}_V$ are to $\mathbf{X}$ and the outputs of each head is summed), layer normalization (either in the Post-LN format (Vaswani et al., 2017; Wang et al., 2019b; Xiong et al., 2020) e.g., for BERT (Kenton & Toutanova, 2019), RoBERTa (Liu et al., 2019), and ALBERT (Lan et al., 2019), or in the Pre-LN format (Baevski & Auli, 2018), e.g., for GPT (Brown et al., 2020), ViT (Dosovitskiy et al., 2021), and PALM (Chowdhery et al., 2023) architectures), and fully-connected layers. Unfortunately, these layers make eigenspectrum analysis intractable (we detail why this is the case in Section 3). However, recent work has demonstrated that simplified Transformer models have surprisingly similar behaviors as full models (Von Oswald et al., 2023; Mahankali et al., 2023; Ahn et al., 2024; Zhang et al., 2024; Ahn et al., 2023). We find that this is also the case for oversmoothing: even though our analysis considers the restricted update in eq. (2) it can explain the smoothing behavior of full Transformer models (e.g., ViT and DeiT models).

### 2.2 What Is Oversmoothing?

In deep learning, 'oversmoothing' broadly describes the tendency of a model to produce more and more similar features as depth increases. For Transformers, prior work largely uses one of three different ways to measure oversmoothing: (a) **Input Convergence**: Do the inputs converge to the exact same feature vector? (Park & Kim, 2022; Wang et al., 2022; Bai et al., 2022); (b) **Angle Convergence**: Do the angles between inputs converge to 0? (Tang et al., 2021; Zhou et al., 2021a; Gong et al., 2021; Yan et al., 2022; Shi et al., 2022; Noci et al., 2022; Guo et al., 2023); (c) **Rank Collapse**: Does the rank of inputs collapse to 1? (Dong et al., 2021; Shi et al., 2022; Noci et al., 2022; Guo et al., 2023; Ali et al., 2023).

**Input Convergence.** One way to formalize oversmoothing is through the lens of signal-processing (Wang et al., 2022): the smoothing of a function can be measured by how much it suppresses higher frequencies in the signal, removing smaller fluctuations to highlight the larger trend. To measure the smoothing of the Transformer update in eq. (2) we can compute the ratio of high frequency signals to low frequency signals preserved in $\mathbf{X}_\ell$. If this goes to 0 as $\ell \to \infty$, all high frequency information is lost: the signal is maximally smoothed. To estimate these signals we can compute the Discrete Fourier Transform (DFT) $\mathcal{F}$ of $\mathbf{X}_\ell$, via $\mathcal{F}(\mathbf{X}_\ell) := \mathbf{F}\mathbf{X}_\ell$, where $\mathbf{F} \in \mathbb{C}^{n \times n}$ is equal to $\mathbf{F}_{k,l} := e^{2\pi \mathrm{i}(k-1)(l-1)}$ for all $k, l \in \{2, \ldots, n\}$ (where $\mathrm{i} := \sqrt{-1}$),

and is 1 otherwise (i.e., in the first row and column). Define the Low Frequency Component (LFC) of $\mathbf{X}_\ell$ as $\mathrm{LFC}[\mathbf{X}_\ell] := \mathbf{F}^{-1}\mathrm{diag}([1,0,\ldots,0])\mathbf{F}\mathbf{X}_\ell = (1/n)\mathbf{1}\mathbf{1}^\top\mathbf{X}_\ell$ (where $\mathbf{1}$ is the column vector of all ones). Further, define the High Frequency Component (HFC) of $\mathbf{X}_\ell$ as $\mathrm{HFC}[\mathbf{X}_\ell] := \mathbf{F}^{-1}\mathrm{diag}([0,1,\ldots,1])\mathbf{F}\mathbf{X}_\ell = (\mathbf{I} - (1/n)\mathbf{1}\mathbf{1}^\top)\mathbf{X}_\ell$. We can now state the first definition of oversmoothing:

**Definition 1** (Input Convergence (Wang et al., 2022))**.** *The Transformer update in eq. (2) oversmooths if for all $\mathbf{X} \in \mathbb{R}^{n \times d}$ we have that*

$$\lim_{\ell \to \infty} \frac{\|\mathrm{HFC}[\mathbf{X}_\ell]\|_2}{\|\mathrm{LFC}[\mathbf{X}_\ell]\|_2} = 0.$$

This definition measures the extent to which inputs converge to the same feature vector. To see this, notice that the term in the numerator $\mathrm{HFC}[\mathbf{X}_\ell] = (\mathbf{I} - (1/n)\mathbf{1}\mathbf{1}^\top)\mathbf{X}_\ell$ goes to 0 if $\mathbf{X}_\ell = \mathbf{1}\overline{\mathbf{x}}^\top$ where $\overline{\mathbf{x}} \in \mathbb{R}^d$ is a vector where entry $\overline{x}_i$ is the mean of the $i$th column of $\mathbf{X}$. This is because $(1/n)\mathbf{1}\mathbf{1}^\top\mathbf{X} = \mathbf{1}\overline{\mathbf{x}}^\top$. Finally, the required condition $\mathbf{X}_\ell = \mathbf{1}\overline{\mathbf{x}}^\top$ only holds when all input vectors are equal. In the following we will refer to the ratio in the above definition as HFC/LFC.

**Angle Convergence.** Another way to quantify oversmoothing is via the cosine similarity between inputs:

**Definition 2** (Angle Convergence)**.** *The Transformer update in eq. (2) oversmooths if for all $\mathbf{X} \in \mathbb{R}^{n \times d}$ we have that*

$$\lim_{\ell \to \infty} \frac{2}{n(n-1)} \sum_{i=1}^{n} \sum_{j=i+1}^{n} \frac{\mathbf{x}_{i,\ell}^\top \mathbf{x}_{j,\ell}}{\|\mathbf{x}_{i,\ell}\|_2 \|\mathbf{x}_{j,\ell}\|_2} = 1,$$

where $\mathbf{x}_{i,\ell} \in \mathbb{R}^d$ is the $i$th row of $\mathbf{X}_\ell$. This measures the cosine of the angle $\theta$ between every pair of inputs $\mathbf{x}_{i,\ell}, \mathbf{x}_{j,\ell}$ and is 1 iff $\theta = 0$.

**Rank Collapse.** Finally, we can also measure oversmoothing via rank collapse in $\mathbf{X}_\ell$. This is usually described as $\lim_{\ell \to \infty} \mathrm{rank}(\mathbf{X}_\ell) = 1$. While rank can be computed via a singular value decomposition (SVD), it is highly-sensitive to the threshold deciding when a singular should be treated as zero. Instead, Guo et al. (2023) use a continuous approximation of rank called the 'effective rank', first introduced by Roy & Vetterli (2007).

**Definition 3** (**Rank Collapse**)**.** *Given $\mathbf{X}_\ell \in \mathbb{R}^{n \times d}$, let $\mathbf{X}_\ell = \mathbf{U}_\ell \mathbf{\Sigma}_\ell \mathbf{V}_\ell$ be a singular value decomposition of $\mathbf{X}$ with singular values $\mathsf{diag}(\mathbf{\Sigma}_\ell) = [\sigma_{1,\ell}, \ldots, \sigma_{r,\ell}]$ for $r \leq \min\{n,d\}$ and $\sigma_{1,\ell} \geq \cdots \geq \sigma_{r,\ell} \geq 0$. Define the following discrete distribution according to the singular values as $p_{i,\ell} = \sigma_{i,\ell} / \sum_{j=1}^{r} \sigma_{j,\ell}$. The effective rank (Roy & Vetterli, 2007) is the exponential of the entropy of this distribution: $\exp(-\sum_{i=1}^{r} -p_{i,\ell} \log p_{i,\ell})$. The Transformer update in eq. (2) oversmooths if for all $\mathbf{X} \in \mathbb{R}^{n \times d}$ we have that*

$$\lim_{\ell \to \infty} \exp(-\sum_{i=1}^{r} p_{i,\ell} \log p_{i,\ell}) = 1.$$

Roy & Vetterli (2007) prove that $1 \leq \exp(-\sum_{i=1}^{r} p_{i,\ell} \log p_{i,\ell}) \leq r$, where $r$ is the rank of $\mathbf{X}_\ell$.

Notice that Definitions 1-3 are progressively relaxed, i.e., if an update satisfies an oversmoothing definition, it also satisfies any later definitions. For each measure, we say that a model producing $\mathbf{X}_\ell$ causes smoothing if the measure approaches the value in each Definition (i.e., towards 0 for Definition 1 and 1 for Definitions 2 & 3). Alternatively, if the measures move away from the value in each Definition, then we say that the model causes sharpening (i.e., Definitions 1 & 3 grow towards $\infty$ and Definition 2 shrinks towards 0).

## 2.3 Observations of Transformer Oversmoothing

The term 'oversmoothing' was first coined by Li et al. (2018) to describe how GNN node features become more similar with more rounds of message passing. A similar observation was made for Transformers by Zhou et al. (2021a). They observed that as depth was increased, the cosine similarity among self-attention

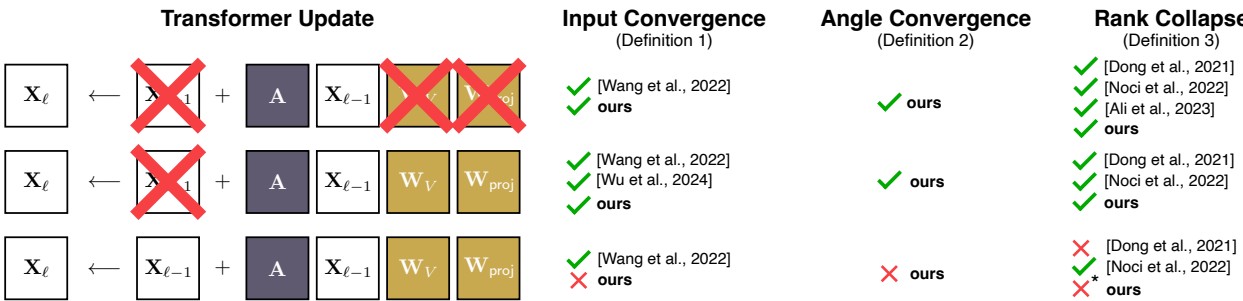

Figure 1: **Theory of Transformer Oversmoothing.** A ✔ indicates prior work says that the corresponding Definition is always satisfied, an ✘ indicates it is not always satisfied. Note that if a Definition is satisfied, then all later Definitions, which are progressively more relaxed, must also be satisfied. The asterisk at the bottom right indicates that Definition 3 is not guaranteed, but it is highly likely.

layers also increased. After this work many other works noticed that feature similarity in vision and language Transformers also increased with depth (Zhou et al., 2021b; Gong et al., 2021; Tang et al., 2021; Raghu et al., 2021; Yan et al., 2022; Shi et al., 2022; Wang et al., 2022; Park & Kim, 2022; Bai et al., 2022; Choi et al., 2023) found. Multiple works around this time found that it was possible to improve vision Transformers by replacing self-attention layers with convolutional layers (Han et al., 2021; Liu et al., 2021b; Jiang et al., 2021; Touvron et al., 2021b; Yuan et al., 2021; Park & Kim, 2022).

## 2.4 The Theory of Transformer Oversmoothing

Figure 1 shows current work on the theory of Transformer oversmoothing for three Transformer updates.

**Input Convergence.** Wang et al. (2022) analyzed oversmoothing from the lens of signal processing (Definition 1). They showed that as the number of self-attention operations tended to infinity, all inputs converge to the same feature vector, producing a low-pass filter. They also analyzed the convergence rate when the residual connection, weights, multiple heads, and a linear layer is added, and found that convergence is not guaranteed. However, they argued that even with these additions oversmoothing still happens: 'it is inevitable that high-frequency components are continuously diluted as ViT goes deeper', i.e., Definition 1 holds. At the same time Shi et al. (2022) investigated oversmoothing using a different notion of input convergence. Curiously, while they argue that oversmoothing can be due to the parameters of layer normalization, their analysis seems to suggest that without layer normalization, oversmoothing does not occur. Recently, Wu et al. (2024) analyze oversmoothing using a metric that is the numerator of the metric defined in Wang et al. (2022). They analyze the transformer update with layer-specific attention and weight matrices and layer normalization, but without the residual connection. Contrary to Shi et al. (2022) they show that layer normalization can prevent oversmoothing. We use the definition of input convergence described by Wang et al. (2022) (our results also hold for the definition used in Wu et al. (2024)).

**Angle Convergence.** As far as we are aware, there is no prior work that directly analyzes oversmoothing from the perspective of angle convergence (Definition 2). However, if an update is shown to input-converge it will also angle-converge (and rank collapse), because input convergence is a stricter requirement than angle convergence (and rank collapse).

**Rank Convergence.** The first work we are aware of that developed a theory around Transformer oversmoothing was Dong et al. (2021) using the notion of rank collapse. Initially, they showed that, without skip-connections, repeated self-attention layers converge double-exponentially to a rank 1 matrix. They then show that there exist models where skip-connections counteract this convergence. Noci et al. (2022) contradict this, arguing that oversmoothing still happens when the residual connection is added, but this can be counteracted if the residual connection is scaled appropriately. Ali et al. (2023) show that rank collapse

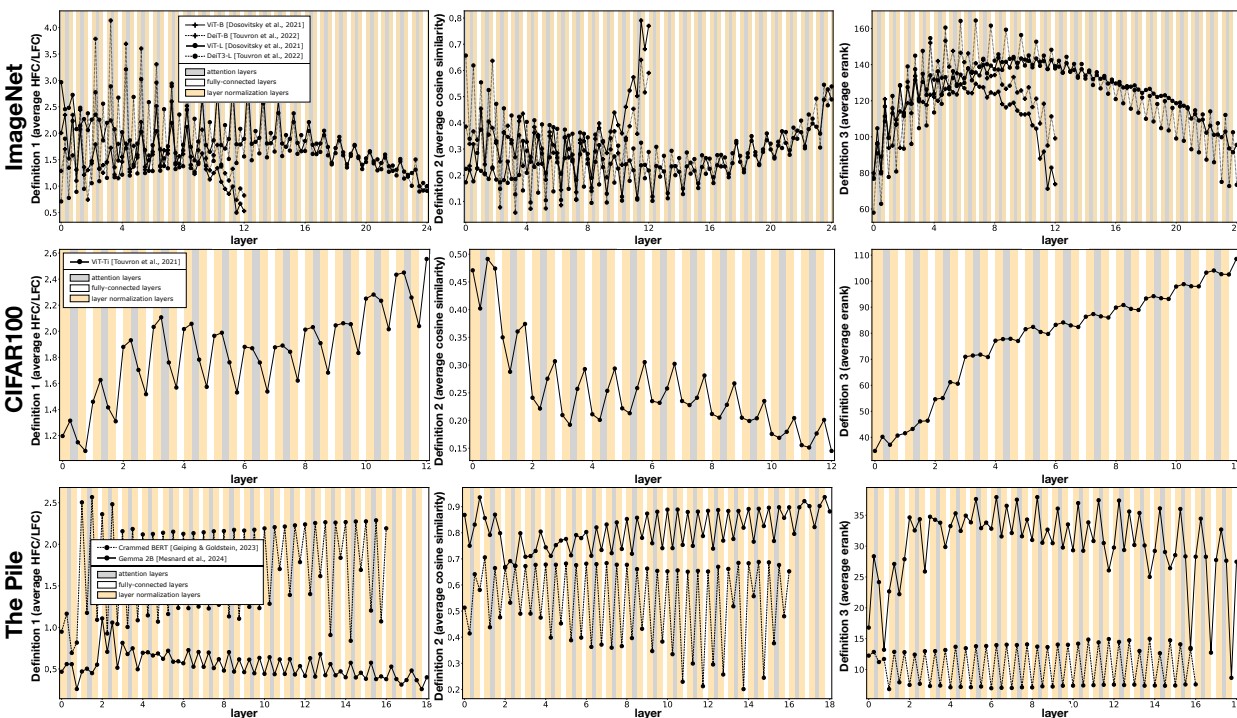

Figure 2: **Are models oversmoothing?** The smoothing metrics defined in Definitions 1 (*left* column), 2 (*center* column), and 3 (*right* column) vs. layer number, for different models and datasets in vision and NLP. Each curve represents a different model. We compute each smoothing metric after each attention layer (grey bars), fully-connect layers (white bars), and layer normalization layers (yellow bars). If oversmoothing was fundamental issue for Transformer models we would expect all curves in the *left* and *right* columns to decrease to 0 and 1, respectively, and all curves in the *center* column to increase to 1. While ImageNet models and Gemma 2B broadly trend these ways, increasing model depth does not exacerbate smoothing, as predicted by prior work. Further, ViT-Ti (Touvron et al., 2021a) on CIFAR100 exhibits the exact opposite trend: smoothing is actually *reduced* by the model. These observations motivate our investigation.

happens without a residual connection and value and projection weights. When a residual connection is added our analysis shows that it is possible to avoid rank collapse.

## 3    Do Transformers Always Oversmooth?

Given the current theory on Transformer oversmoothing, how are Transformer models so successful for vision and NLP applications (Kenton & Toutanova, 2019; Liu et al., 2019; Lan et al., 2019; Brown et al., 2020; Dosovitskiy et al., 2021; Chowdhery et al., 2023)? To investigate this, we computed the above three metrics in Definitions 1-3 on a set of pre-trained models for vision and NLP that have been used in prior work on oversmoothing (Wang et al., 2022; Choi et al., 2023) in Figure 2. We notice that for all ImageNet models (ViT-B, ViT-L (Dosovitskiy et al., 2021), DeiT-B (Touvron et al., 2021a), DeiT3-L (Touvron et al., 2022)), as depth increases, we do see the metrics approaching their oversmoothing values as described in Definitions 1-3. Rank (Definition 3) does not consistently decrease and stays relatively high for 12 layer models, but continues to drop as depth is increased. However, we see something completely unexpected from the CIFAR model (ViT-Ti (Touvron et al., 2021a)). All of the metrics *show reduction in smoothing behavior* as depth increases. Similarly, for The Pile model (Crammed BERT (Geiping & Goldstein, 2023)) we see behavior that appears to oscillate between more and less smoothing. These behaviors motivate us to further investigate the Transformer update.

### 3.1 Preliminaries

Our strategy will be to understand the eigenspectrum of the Transformer update in the limit and to use this understanding to derive what the features $\mathbf{X}_\ell$ converge to as $\ell \to \infty$. This will allow us to understand if and when Definitions 1-3 hold. We start by rewriting the Transformer update, eq. (2), to make it more amenable to analysis. Define the $\text{vec}(\mathbf{M})$ operator as converting any matrix $\mathbf{M}$ to a vector $\mathbf{m}$ by stacking its columns. We can rewrite eq. (2) vectorized as follows

$$\text{vec}(\mathbf{X}_\ell) = (\mathbf{I} + \underbrace{\mathbf{W}_{\text{proj}}^\top \mathbf{W}_V^\top}_{:=\mathbf{H}} \otimes \mathbf{A})\text{vec}(\mathbf{X}_{\ell-1}). \tag{3}$$

This formulation is especially useful because $\text{vec}(\mathbf{X}_\ell) = (\mathbf{I} + \mathbf{H} \otimes \mathbf{A})^\ell \text{vec}(\mathbf{X}_0)$. We now introduce an assumption on $\mathbf{A}$ that is also used in prior work (Ali et al., 2023; Wang et al., 2022).

**Assumption 1** (Ali et al. (2023); Wang et al. (2022)). *The attention matrix is positive, i.e., $\mathbf{A} > 0$, and diagonalizable.*

This assumption nearly always holds unless $\mathbf{A}$ numerically underflows. In our experiments we never encountered $a_{ij} = 0$ for any element $(i, j) \in \mathbb{R}^n \times \mathbb{R}^n$ or $\mathbf{A}$ that was not diagonalizable, in any architecture. Note $\mathbf{A}$ is also right-stochastic, i.e., $\sum_j a_{i,j} = 1$, by definition in eq. (1). This combined with Assumption 1 immediately implies the following proposition.

**Proposition 1** (Meyer & Stewart (2023)). *Given Assumption 1, all eigenvalues of $\mathbf{A}$ lie within $(-1, 1]$. There is one largest eigenvalue that is equal to 1, with corresponding unique eigenvector $\mathbf{1}$.*

We leave the proof to the Appendix. We can now analyze the eigenvalues of the Transformer update equations.

### 3.2 The Eigenvalues

First notice that the eigenvalues of $(\mathbf{I} + \mathbf{H} \otimes \mathbf{A})^\ell$ can be written in terms of the eigenvalues of $\mathbf{H}, \mathbf{A}$:

**Lemma 1.** *Let $\lambda_1^A, \ldots, \lambda_n^A$ be the eigenvalues of $\mathbf{A}$ and let $\lambda_1^H, \ldots, \lambda_d^H$ be the eigenvalues of $\mathbf{H}$. The eigenvalues of $(\mathbf{I} + \mathbf{H} \otimes \mathbf{A})^\ell$ are equal to $(1 + \lambda_j^H \lambda_i^A)$ for $j \in \{1, \ldots, d\}$ and $i \in \{1, \ldots, n\}$.*

The proof can be derived from Theorem 2.3 of Schacke (2004). Given this, notice that as the number of layers $\ell$ in the Transformer update eq. (3) increases, one eigenvalue $(1 + \lambda_{j^*}^H \lambda_{i^*}^A)$ will dominate the rest (except in cases of ties).

**Definition 4** (Dominating eigenvalue(s)). *At least one of the eigenvalues of $(\mathbf{I} + \mathbf{H} \otimes \mathbf{A})$ has a larger magnitude than all others, i.e., there exists $j^*, i^*$ (which may be a set of indices if there are ties) such that $|1 + \lambda_{j^*}^H \lambda_{i^*}^A| > |1 + \lambda_{j'}^H \lambda_{i'}^A|$ for all $j' \in \{1, \ldots, d\} \setminus j^*$ and $i' \in \{1, \ldots, n\} \setminus i^*$. These eigenvalues are called **dominating**.*

Which eigenvalue dominates will control the smoothing behavior of the Transformer.

**Theorem 1.** *Given the Transformer update in eq. (3), let $\{\lambda_i^A\}_{i=1}^n$ and $\{\lambda_j^H\}_{j=1}^d$ be the eigenvalues of $\mathbf{A}$ and $\mathbf{H}$. Let the eigenvalues be sorted as follows, $\lambda_1^A \leq \cdots \leq \lambda_n^A$ and $|1 + \lambda_1^H| \leq \cdots \leq |1 + \lambda_d^H|$. As the number of layers $\ell \to \infty$, there are two types of dominating eigenvalues: (1) $(1 + \lambda_{j^*}^H \lambda_n^A)$. and (2) $(1 + \lambda_{j^*}^H \lambda_1^A)$*

We leave the proof to the Appendix (where we describe all possible cases). We can now use this result to derive what $\mathbf{X}_\ell$ converges to as depth increases.

### 3.3 The Features

**Theorem 2.** *Given the Transformer update in eq. (3), if a single eigenvalue dominates, as the number of total layers $\ell \to \infty$, the feature representation $\mathbf{X}_\ell$ converges to one of two representations: (1) If $(1 + \lambda_j^H \lambda_n^A)$ dominates then[1],*

$$\mathbf{X}_\ell \to (1 + \lambda_j^H \lambda_n^A)^\ell s_{j,n} \mathbf{1} \mathbf{v}_j^{H\top}, \tag{4}$$

---

[1] From Proposition 1 we have that, $\mathbf{v}_n^A = \mathbf{1}$.

**(2)** If $(1 + \lambda_j^H \lambda_1^A)$ *dominates then,*

$$\mathbf{X}_\ell \to (1 + \lambda_j^H \lambda_1^A)^\ell s_{j,1} \mathbf{v}_1^A \mathbf{v}_j^{H\top} \tag{5}$$

*where* $\mathbf{v}^H, \mathbf{v}^A$ *are eigenvalues of* $\mathbf{H}, \mathbf{A}$ *and* $s_{j,i} := \langle \mathbf{v}_{j,i}^{Q^{-1}}, \mathsf{vec}(\mathbf{X}) \rangle$ *and* $\mathbf{v}_{j,i}^{Q^{-1}}$ *is row* $ji$ *in the matrix* $\mathbf{Q}^{-1}$ *(here* $\mathbf{Q}$ *is the matrix of eigenvectors of* $(\mathbf{I} + \mathbf{H} \otimes \mathbf{A})$*).* **(3)** *If multiple eigenvalues have the same dominating magnitude,* $\mathbf{X}_\ell$ *converges to the sum of the dominating terms.*

**Corollary 1.** *If the residual connection is removed in the Transformer update, then the eigenvalues are of the form* $(\lambda_j^H \lambda_i^A)$. *Further,* $(\lambda_{j*}^H \lambda_n^A)$ *is always a dominating eigenvalue, and* $\mathbf{X}_\ell \to$ $\mathbf{1}\big(\sum_{j* \in \mathcal{E}_{\max}^H} (\lambda_{j*}^H \lambda_n^A)^\ell s_{j*,n} \mathbf{v}_{j*}^{H\top}\big)$ *as* $\ell \to \infty$*, where* $\mathcal{E}_{\max}^H$ *is the set of all eigenvalue indices equal to the dominating eigenvalue* $\lambda_{j*}^H$*.*

See the Appendix for proofs of the above statements. Given these results, we can now understand when the oversmoothing definitions apply.

### 3.4 When Oversmoothing Happens

**Theorem 3.** *Given the Transformer update eq. (3), as the number of total layers* $\ell \to \infty$*, if* **(1)** *one eigenvalue* $(1 + \lambda_{j*}^H \lambda_n^A)$ *dominates, we have input convergence, angle convergence, and rank collapse. If* **(2)** *one eigenvalue* $(1 + \lambda_{j*}^H \lambda_1^A)$ *dominates, we do not have input convergence or angle convergence, but we do have rank collapse. If* **(3)** *multiple eigenvalues have the same dominating magnitude and: (a) there is at least one dominating eigenvalue* $(1 + \lambda_{j*}^H \lambda_{i*}^A)$ *where* $\lambda_{i*}^A \neq \lambda_n^A$*, then we do not have input convergence or angle convergence, if also (b) the geometric multiplicity of* $\lambda_1^A$ *and* $\lambda_{j*}^H$ *are both greater than 1, then we also do not have rank collapse.*

**Corollary 2.** *If the residual connection is removed in the Transformer update, input convergence, angle convergence, and rank collapse are guaranteed.*

The proofs are left to the Appendix. The above statements follow directly from Theorem 2 and Corollary 1. They tell us that whenever a single eigenvalue $(1 + \lambda_j^H \lambda_n^A)$ dominates, *every input in* $\mathbf{X}_\ell$ *converges to the same feature vector*. This happens because $\mathbf{v}_n^A = \mathbf{1}$ and so $\mathbf{x}_{\ell,i} \sim \mathbf{v}_j^H$, for all $i$ as $\ell \to \infty$. But there is a second case: whenever the single eigenvalue $(1 + \lambda_j^H \lambda_1^A)$ dominates, each feature is not guaranteed to be identical. However, $\mathbf{X}_\ell \to (1 + \lambda_j^H \lambda_1^A)^\ell s_{j,1} \mathbf{v}_1^A \mathbf{v}_j^{H\top}$ is still a matrix of rank one. If instead multiple eigenvalue dominate and the geometric multiplicity of $\lambda_1^A$ and $\lambda_{j*}^H$ are both greater than 1 then $\mathbf{X}_\ell$ is a sum of at least 2 rank-1 matrices and so we do not have rank collapse.

Theorem 3 largely contradicts prior theoretical results on oversmoothing. We suspect a few reasons for this. First, if multiple types of analyses are used within one paper, and they give conflicting results, resolving this can be especially challenging (Wang et al., 2022). Second, certain assumptions may not always hold in practice, e.g., Noci et al. (2022) assume that $\mathbf{A} = \frac{1}{n} \mathbf{1}\mathbf{1}^\top$ at initialization.

**On Layer Normalization & Feed Forward Layers.** Most Transformers also include layer normalization and feedforward layers. Unfortunately, both of these break our analysis. For instance, a repeated Pre-LN layer can be represented by the following update,

$$\mathsf{vec}(\mathbf{X}_\ell) = (\mathbf{I} + \mathbf{H}\mathbf{D}^{-1} \otimes \mathbf{A})^\ell \mathsf{vec}(\mathbf{X}_0) - \ell(\mathsf{vec}(\mathbf{A}\mathbf{1}\mathbf{b}^\top \mathbf{D}^{-1}\mathbf{H}^\top)),$$

where $\mathbf{b}$ and $\mathbf{D}^{-1}$ are terms introduced by the normalization layer. However, as far as we are aware there is no way to characterize the relationship between the eigenvalues of $(\mathbf{I} + \mathbf{H}\mathbf{D}^{-1} \otimes \mathbf{A})$ and the eigenvalues of $\mathbf{H}$, $\mathbf{A}$, and $\mathbf{D}$, without introducing further assumptions (e.g., if $\mathbf{H}$ is symmetric there is a known relationship). This difficulty also applies to Post-LN layers. We encounter a similar difficulty for feed forward layers,

$$\mathsf{vec}(\mathbf{X}_\ell) = (\mathbf{W}^\top \otimes \mathbf{I} + \mathbf{W}^\top \mathbf{H} \otimes \mathbf{A})^\ell \mathsf{vec}(\mathbf{X}_0),$$

where $\mathbf{W}$ is the parameter of the feed forward layer. Similar to layer normalization, as far as we are aware, we cannot characterize the eigenvalues of $(\mathbf{W}^\top \otimes \mathbf{I} + \mathbf{W}^\top \mathbf{H} \otimes \mathbf{A})$ in terms of the eigenvalues of $\mathbf{H}$, $\mathbf{A}$, and $\mathbf{W}$, without further assumptions.

> **Takeaways:** We find that oversmoothing is not inevitable. Whether a layer smooths its input depends on the interaction between the spectrum of the attention $\mathbf{A}$ and weights $\mathbf{H}$. The largest eigenvalue of $(\mathbf{I} + \mathbf{H} \otimes \mathbf{A})$ dictates smoothing behaviour: if $(1 + \lambda_j^H \lambda_n^A)$ is largest, the inputs eventually converge to the same feature vector, but if $(1 + \lambda_j^H \lambda_1^A)$ is largest, they do not.

A natural question is can we use the above analysis to influence the smoothing behavior of Transformer models? In the next section we derive a Corollary of Theorem 1 that allows one to do so using a simple reparameterization of $\mathbf{H}$.

## 4 Testing the Theory

How applicable are the theoretical results developed in the previous section? Similar to recent theoretical work on Transformer optimization (Ahn et al., 2023; Mahankali et al., 2023; Von Oswald et al., 2023; Ahn et al., 2024; Zhang et al., 2024), the Transformer update we analyze in eq. (3) is simplified: no positional encoding, fixed attention and weights, single-head attention, no layer normalization or feed-forward layers. How we understand the explanatory impact of our theory on full-scale Transformer models? To do so, we derive a simple reparameterization of the weights $\mathbf{H}$ that allows one influence smoothing behavior. We can then test this parameterization in existing Transformer architectures to see if smoothing can be affected, and also judge its impact on generalization.

To derive this reparameterization, first note that Theorem 3 tells us that if $(1 + \lambda_j^H \lambda_n^A)$ dominates then this will cause oversmoothing, whereas if instead $(1 + \lambda_j^H \lambda_1^A)$ dominates we avoid it. To find $\mathbf{A}$ and $\mathbf{H}$ that are guaranteed to have either $(1 + \lambda_j^H \lambda_1^A)$ or $(1 + \lambda_j^H \lambda_n^A)$ dominate we could dig through the proof of Theorem 1 and consider all cases. However, as $\mathbf{A}$ changes for every batch of data $\mathbf{X}$ there is no easy way to guarantee the smoothing behavior of a model. Because of this, we need a solution that involves only controlling the eigenvalues of $\mathbf{H}$. Luckily, we can simplify the proof of Theorem 1 into a much simpler condition.

**Corollary 3.** *If the eigenvalues of $\mathbf{H}$ fall within $[-1, 0)$, then $(1 + \lambda_{j*}^H \lambda_1^A)$ dominates. If the eigenvalues of $\mathbf{H}$ fall within $(0, \infty)$, then $(1 + \lambda_{j*}^H \lambda_n^A)$ dominates.*

See the Appendix for a proof. To ensure that the eigenvalues of $\mathbf{H}$ fall in these ranges, we propose to directly parameterize its eigendecomposition. Specifically, define $\mathbf{H}$ as $\mathbf{H} = \mathbf{V}_H \Lambda_H \mathbf{V}_H^{-1}$, where $\mathbf{V}_H$ is a full-rank matrix and $\Lambda_H$ is diagonal. We learn parameters $\mathbf{V}_H$ by taking gradients in the standard way (i.e., directly and through the inversion). To learn the diagonal of $\Lambda_H$, i.e., $\mathsf{diag}(\Lambda_H)$, we parameterize the sharpening model as $\mathsf{diag}(\Lambda_H) := \mathsf{clip}(\psi, [-1, 0])$, where $\psi$ are tunable parameters and $\mathsf{clip}(\psi, [l, u]) := \min(\max(\psi, l), u)$ forces all of $\psi$ to lie in $[l, u]$. Similarly we parameterize the smoothing model as $\mathsf{diag}(\Lambda_H) := \mathsf{clip}(\psi, [0, 1])$.[2]

## 5 Experiments

We now run experiments using our reparameterization, in order to understand the predictive power of our results on full Transformer architectures. We train sharpening and smoothing models on CIFAR100 (Krizhevsky et al., 2009), ImageNet (Deng et al., 2009), and The Pile (Gao et al., 2020). We expect that the attention layers of sharpening models reduce smoothing, i.e., HFC/LFC (Definition 1) increases, cosine similarity (Definition 2) decreases, and effective rank (Definition 3) increases, while smoothing attention layers increase smoothing.

**Initialization.** We initialize $\mathbf{H} = \mathbf{V}_H \Lambda_H \mathbf{V}_H^{-1}$ to mimic the initializations used in the ViT-Ti and Bert baselines, which are initialized using He initialization (He et al., 2015). Specifically, we first initialize $\mathbf{V}_H$

---

[2]While we could have allowed the smoothing model to use the space of positive reals via $\mathsf{diag}(\Lambda_H) := |\psi|$, we found that restricting the space of allowed eigenvalues stabilized training.

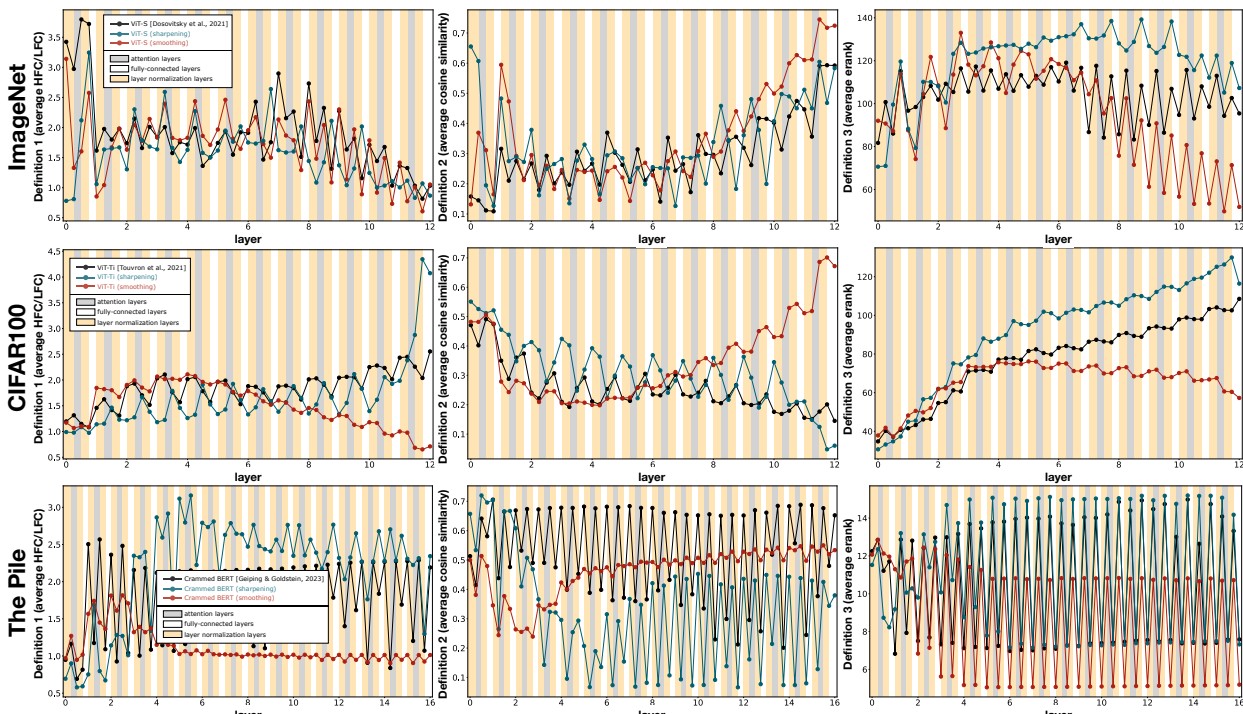

Figure 3: **Testing the theory.** We take a model architecture for each dataset and reparameterize $\mathbf{H}$ as $\mathbf{H} = \mathbf{V}_H \Lambda_H \mathbf{V}_H^{-1}$. We then train a *sharpening* model where $\mathsf{diag}(\Lambda_H) := \mathsf{clip}(\psi, [-1, 0])$ and a *smoothing* model where $\mathsf{diag}(\Lambda_H) := \mathsf{clip}(\psi, [0, 1])$. Here $\psi$ are tunable parameters. Our theory suggests that the *smoothing* model should cause the metric of Definition 1 (*left* column) to decrease, Definition 2 (*center* column) to increase, and Definition 3 (*right* column) to decrease. These trends seem to generally hold for CIFAR100 and The Pile, but not for ImageNet. We suspect this has to do with impact of layer normalization, which we test in Figures 4 and 6.

|  | CIFAR100 | | | ImageNet | | | The Pile | | |
|---|---|---|---|---|---|---|---|---|---|
|  | ViT-Ti | ViT-Ti | ViT-Ti | ViT-S | ViT-S | ViT-S | Cram. Bert | Cram. Bert | Cram. Bert |
| Attention | 0.000 | 1.000 | 0.083 | 0.000 | 0.333 | 0.167 | 0.667 | 0.467 | 0.200 |
| LayerNorm | 0.500 | 0.542 | 0.125 | 0.833 | 0.708 | 0.542 | 0.750 | 0.750 | 0.156 |
| MLP | 1.000 | 0.167 | 1.000 | 0.583 | 0.167 | 0.500 | 0.600 | 0.600 | 0.267 |

Table 1: Proportion of layers that increase the HFC/LFC (Definition 1) for each layer type.

|  | CIFAR100 | | | ImageNet | | | The Pile | | |
|---|---|---|---|---|---|---|---|---|---|
|  | ViT-Ti | ViT-Ti | ViT-Ti | ViT-S | ViT-S | ViT-S | Cram. Bert | Cram. Bert | Cram. Bert |
| Attention | 1.000 | 0.000 | 1.000 | 1.000 | 0.667 | 0.833 | 0.333 | 0.533 | 0.800 |
| LayerNorm | 0.458 | 0.458 | 0.708 | 0.167 | 0.250 | 0.458 | 0.469 | 0.250 | 0.688 |
| MLP | 0.000 | 0.917 | 0.000 | 0.417 | 0.833 | 0.583 | 0.333 | 0.400 | 0.733 |

Table 2: Proportion of layers that increase the cosine similarity (Definition 2) for each layer type.

using He initialization. To initialize $\mathsf{diag}(\Lambda_H)$ we sample from a normal distribution with mean 0, and variance 0.1 (we show the impact of different initialization variances on smoothing and training loss in Figures 8 and 9). All other training and architecture details are in the Appendix.

| | CIFAR100 | | | ImageNet | | | The Pile | | |
|---|---|---|---|---|---|---|---|---|---|
| | ViT-Ti | ViT-Ti | ViT-Ti | ViT-S | ViT-S | ViT-S | Cram. Bert | Cram. Bert | Cram. Bert |
| Attention | 0.250 | 1.000 | 0.167 | 0.083 | 0.833 | 0.500 | 0.733 | 0.600 | 0.267 |
| LayerNorm | 0.583 | 0.667 | 0.583 | 0.875 | 0.500 | 0.583 | 0.594 | 0.594 | 0.781 |
| MLP | 0.917 | 0.417 | 0.917 | 0.417 | 0.250 | 0.167 | 0.733 | 0.733 | 0.333 |

Table 3: Proportion of layers that increase the effective rank (Definition 3) for each layer type.

| | HFC/LFC, 1 | | | cosine similarity, 2 | | | effective rank, 3 | | |
|---|---|---|---|---|---|---|---|---|---|
| | ViT-S | ViT-S | ViT-S | ViT-S | ViT-S | ViT-S | ViT-S | ViT-S | ViT-S |
| Attention | 0.000 | 1.000 | 0.250 | 1.000 | 0.000 | 0.750 | 0.083 | 0.500 | 0.250 |
| LayerNorm | 0.833 | 1.000 | 0.958 | 0.167 | 0.000 | 0.042 | 0.875 | 0.750 | 0.958 |
| MLP | 0.583 | 0.167 | 0.417 | 0.417 | 0.833 | 0.583 | 0.417 | 0.833 | 0.667 |

Table 4: **No LN weights.** Proportion of layers that increase HFC/LFC, cosine similarity, effective rank.

**Reparameterization results.** Figure 3 shows the effect of the two reparameterizations: sharpening and smoothing. Since our analysis deals with the attention layer, we plot the variations in metrics for each part of the attention block to clearly visualize their individual impact. This is useful since those layers can have different, sometimes opposite impacts on the metrics we are interested in.

We immediately notice that the effects of the reparameterizations vary wildly depending on the dataset and the smoothing metric. On one end of the spectrum is CIFAR100: where the sharpening model reduces smoothing and the smoothing model increases smoothing, for all metrics. The theory seems to predict the attention layer perfectly here as 100% of the attention layers in the sharpening model increase the HFC/LFC (Table 1) and the effective rank (Table 3) and 0% of the increase the cosine similarity (Table 2). We observe nearly the opposite, intended, behaviour for the smoothing model. On the other end of the spectrum is ImageNet, where the reparameterizations seem to have little effect on the HFC/LFC and cosine similarity smoothing metrics. Here only 33% of the attention layers of the sharpening model increase the HFC/LFC (Table 1) and a majority of attention layers actually increase the cosine similarity (67%, Table 2), the opposite of its intended effect! The effective rank can be slightly manipulated: 83% of attention layers of the sharpening model increase the effective rank (Table 3). However, the final smoothing of the model is nearly the same as the original model as both LayerNorm and MLP layers of the sharpening model increase the effective rank less than the original model. In between these extremes is The Pile, where the smoothing model seems to predictably reduce the HFC/LFC and effective rank, compared to the original model, dropping the percentage of attention layers that increase HFC/LFC down from 67% to 20% (Table 1), and dropping effective-rank-increasing layers from 73% down to 27% (Table 3). However, the smoothing model has less overall smoothing than the base model, as defined by cosine similarity (Figure 3).

**The impact of layer normalization.** We suspect the reason we observe such varied effects is due to interactions between attention layers and other layers that we were unable to analyze. To test this theory, we run a simple experiment: we create two 'repeating' models that obey eq. (2), i.e., weights and attention repeat instead of being layer-dependent. The first model is reparameterized as above to be sharpening (i.e., $\mathbf{W}_{proj}^\top \mathbf{W}_V^\top :=$ $\mathbf{H} = \mathbf{V}_H \Lambda_H \mathbf{V}_H^{-1}$ such that $\mathsf{diag}(\Lambda_H) := \mathsf{clip}(\psi, [-1, 0])$) and the second is reparameterized to be smoothing. This way, both models follow our theory exactly so, by Corollary 3, sharpening and smoothing is guaranteed. Figure 4 (*left*) shows the HFC/LFC (Definition 1) of each repeating model, and as predicted, the HFC/LFC of the

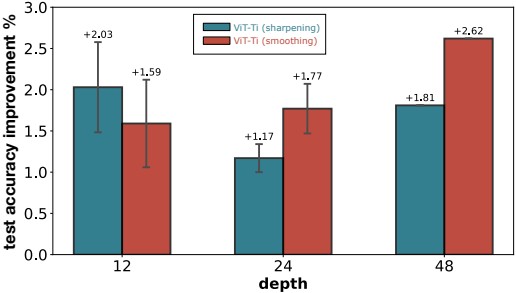

Figure 5: **Test accuracy improvements (CI-FAR100).** Test results of reparameterized models for model depths $\{12, 24, 48\}$. Results are averaged over 3 trials, numbers over bars report average accuracy, gray bars show standard deviations.

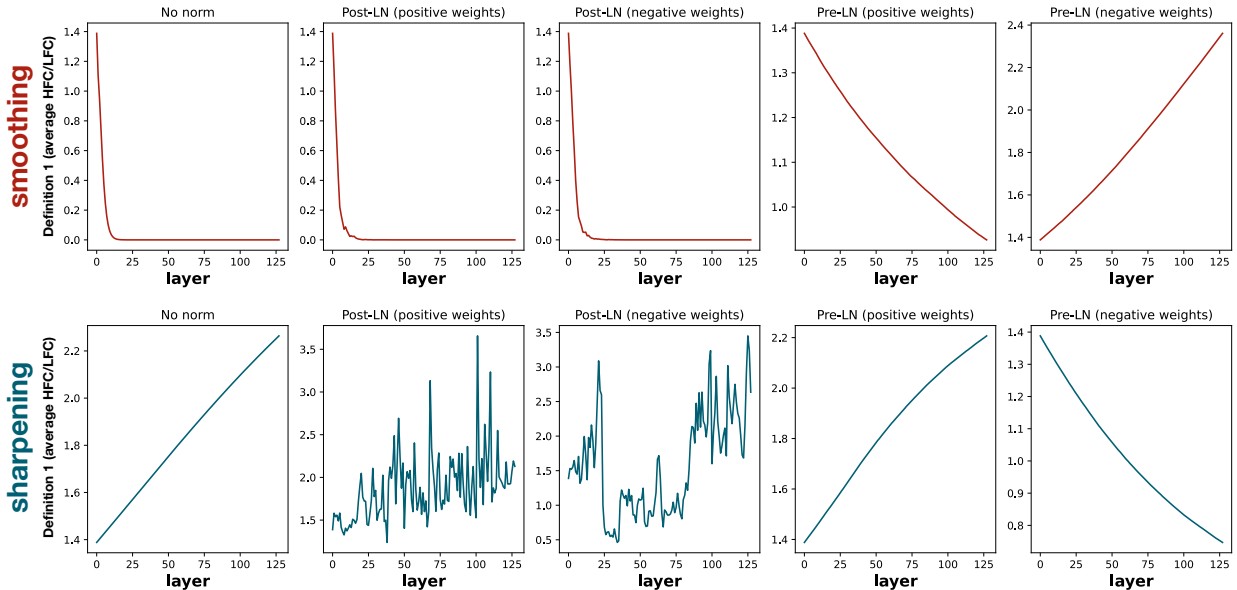

Figure 4: **Impact of Layer Normalization.** The average HFC/LFC for the Transformer update with repeated layers, as in eq. (3), and different types of layer normalization (Post-LN (Vaswani et al., 2017), Pre-LN (Baevski & Auli, 2018)) where the weights of the layer normalization are fixed to be positive or negative. See text for details.

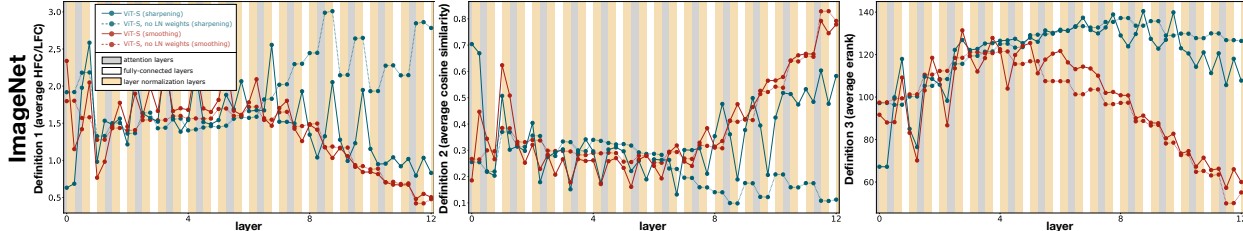

Figure 6: **Smoothing, no LN weights.** HFC/LFC (Definition 1), cosine similarity (Definition 2), and effective rank (Definition 3) for reparameterized models with and without layer normalization weights.

smoothing model decreases (exponentially), approaching 0, while the HFC/LFC of the sharpening increases. We then introduce Post-LN (Vaswani et al., 2017) and Pre-LN (Baevski & Auli, 2018) into the repeating architectures and plot the HFC/LFC when layer normalization weights ($\gamma$) are either positive or negative (both randomly sampled).[3] We observe that Post-LN has no impact on the behaviour of the smoothing model. Post-LN does affect the sharpening model, creating a more erratic sharpening pattern, however the overall trend is preserved. On the other hand, Pre-LN (which is used in all of the full Transformer architectures in our experiments) reverses the expected behaviour when weights are negative. This could explain the surprising results we observe in Figure 3.

To test this in full Transformer models, we train smoothing and sharpening models without layer normalization weights (i.e, we fix $\gamma = 1$, $\beta = 0$) on ImageNet, the dataset most resistant to reparameterization. The results are shown in Figure 6. While the smoothing model behaves similarly to the model with layer normalization weights, the sharpening model without layer normalization weights reduces smoothing far beyond the model with layer normalization weights. This is particularly apparent for HFC/LFC, where the percentage of layers that increase HFC/LFC goes from 33% to 100% (Table 4), and for cosine similarity, where the percentage of layers increasing cosine similarity go from 67% to 0% (Table 4).

---

[3]We omit the bias term $\beta$ to isolate the impact of layer normalization weights.

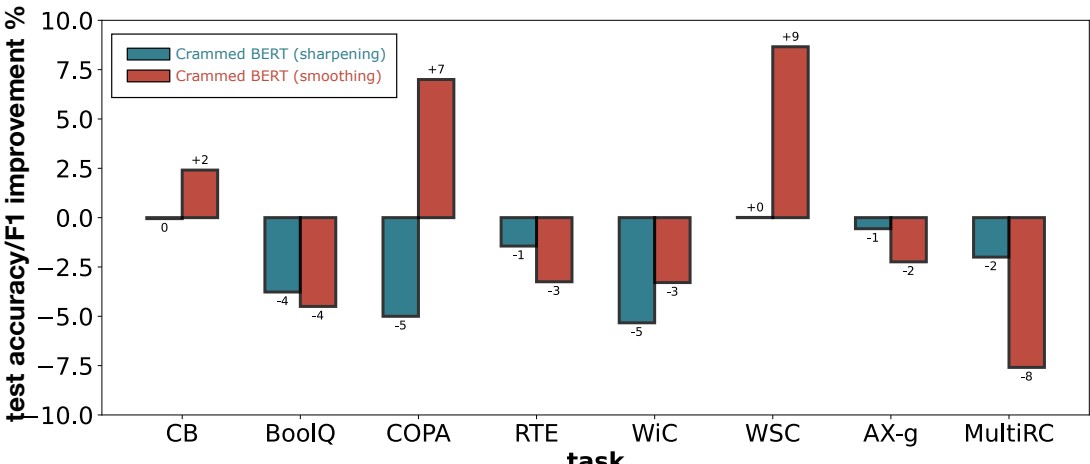

Figure 7: **Test accuracy/F1 improvements (SuperGLUE).** Test results of reparameterized models.

Overall, these results show some evidence that our theoretical analysis can explain the smoothing behaviour of full Transformer models. When it falls short, one possible explanation is the impact of layer normalization weights. When these weights are removed in full Transformer models on ImageNet, our theoretical results can better explain their smoothing behaviour. This explanation is far from perfect: the smoothing models on ImageNet have nearly the same smoothing behaviour as the original model as measured by HFC/LFC or cosine similarity, whether layer normalization weights are included in the model or not. It would be very interesting to extend the analysis to explain this behaviour.

**Test accuracy.** Our goal in this section was to test how well our theory can explain full Transformer architectures. While our intention is not to suggest using the reparameterized sharpening or smoothing models in practice, we were curious to see what impact they had on test accuracy. Recall that prior work argued that the reason why test accuracy flattened as model depth increased was because of oversmoothing. This implies that sharpening models should increase test accuracy beyond the baseline model, whereas smoothing models should reduce it. However, for ImageNet both sharpening and smoothing models have lower test accuracy than the original model: $-2.6\%$ (sharpening) and $-0.6\%$ (smoothing). This seems to contradict the oversmoothing hypothesis. For CIFAR100 (results in Figure 5), sharpening does improve test accuracy, but as model depth is increased, smoothing improves it even more. This is the exact opposite of what is stated by the oversmoothing hypothesis. Finally, we evaluate the reparameterized Crammed BERT (Geiping & Goldstein, 2023) models on SuperGlue (Wang et al., 2019a) (following the literature we report changes in test F1 for the CB and MultiRC tasks, and test accuracy for the rest). Both Crammed Bert (shapening) and Crammed Bert (smoothing) largely harm the performance of the original model. The only benefits come from the smoothing model.

---

**Takeaways:** To understand how much of our theoretical analysis applies to full Transformer models, we attempt to reduce or increase smoothing using sharpening and smoothing models. For certain datasets, these reparameterized models act as predicted by the theory (i.e., CIFAR100). For others, they have little effect (i.e., ImageNet). We hypothesize that this is due to the impact of layer normalization, and we find that LN weights can reverse the intended effects of 'repeated' versions of the sharpening and smoothing models that follow eq. 2. When we remove layer normalization weights in full Transformer models the sharpening model significantly reduces smoothing on ImageNet. We investigate the sensitivity of the reparameterized models with respect to their hyperparameters (see Appendix C). Finally, we report the test accuracy of the reparameterized models, which defy the oversmoothing hypothesis for Transformers.

---

## 6 Discussion

One limitation of the current theoretical analysis is that the results are asymptotic, applying in the limit as $\ell \to \infty$. It would be useful to understand the rates of convergence. We would also like to expand the theoretical analysis to account for layer normalization and fully-connected layers. Special conditions will likely need to be placed on $\mathbf{H}$ to enable this analysis, such as symmetric $\mathbf{A}, \mathbf{H}$ (Sander et al., 2022). Another limitation is that we do not take into account language modeling aspects such as causal attention and positional encoding, which Barbero et al. (2024) relate to oversquashing, another phenomenon discussed in graph neural network literature. Considering how oversmoothing and oversquashing both lead to a form of collapse, it would be interesting to unify these views.

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

## Appendix

## A    Proofs

**Proposition 1** (Meyer & Stewart (2023))**.** *Given Assumption 1, all eigenvalues of* $\mathbf{A}$ *lie within* $(-1, 1]$. *There is one largest eigenvalue that is equal to* $1$, *with corresponding unique eigenvector* $\mathbf{1}$.

*Proof.* First, because $\mathbf{A}$ is positive, by the Perron-Frobenius Theorem Meyer & Stewart (2023) all eigenvalues of $\mathbf{A}$ are in $\mathbb{R}$ (and so there exist associated eigenvectors that are also in $\mathbb{R}$). Next, recall the definition of an eigenvalue $\lambda$ and eigenvector $\mathbf{v}$: $\mathbf{A}\mathbf{v} = \lambda\mathbf{v}$. Let us write the equation for any row $i \in \{1, \ldots, n\}$ explicitly:

$$a_{i1}v_1 + \cdots + a_{in}v_n = \lambda v_i.$$

Further let,

$$v_{\max} := \max\{|v_1|, \ldots, |v_n|\} \tag{6}$$

Note that $v_{\max} > 0$, otherwise it is not a valid eigenvector. Further let $k_{\max}$ be the index of $\mathbf{v}$ corresponding to $v_{\max}$. Then we have,

$$
\begin{aligned}
|\lambda|v_{\max} &= |a_{k_{\max}1}v_1 + \cdots + a_{k_{\max}n}v_n| \\
&\leq a_{k_{\max}1}|v_1| + \cdots + a_{k_{\max}n}|v_n| \\
&\leq a_{k_{\max}1}|v_{k_{\max}}| + \cdots + a_{k_{\max}n}|v_{k_{\max}}| \\
&= (a_{k_{\max}1} + \cdots + a_{k_{\max}n})|v_{k_{\max}}| = |v_{\max}|
\end{aligned}
$$

The first inequality is given by the triangle inequality and because $a_{ij} > 0$. The second is given by the definition of $v_{\max}$ as the maximal element in $\mathbf{v}$. The final inequality is given by the definition of $\mathbf{A}$ in eq. (1) as right stochastic (i.e., all rows of $\mathbf{A}$ sum to 1) and because $|v_{k_{\max}}| = |v_{\max}|$. Next, note that because $v_{\max} > 0$, it must be that $\lambda \leq 1$. Finally, to show that the one largest eigenvalue is equal to 1, recall by the definition of $\mathbf{A}$ in eq. (1) that $\mathbf{A}\mathbf{1} = \mathbf{1}$, where $\mathbf{1}$ is the vector of all ones. So $\mathbf{1}$ is an eigenvector of $\mathbf{A}$, with eigenvalue $\lambda^* = 1$. Because $a_{ij} > 0$, and we showed above that all eigenvalues must lie in in $[-1, 1]$, by the Perron-Frobenius theorem Meyer & Stewart (2023) $\lambda^* = 1$ is the Perron root. This means that all other eigenvalues $\lambda_i$ satisfy the following inequality $|\lambda_i| < \lambda^*$. Further $\mathbf{1}$ is the Perron eigenvector, and all other eigenvectors have at least one negative component, making $\mathbf{1}$ unique. Finally, because $\mathbf{A}$ is diagonalizable it has $n$ linearly independent eigenvectors. $\qquad\square$

We now prove a lemma that will allow us to prove Theorem 1.

**Lemma 2.** *Consider the Transformer update in eq. (3). Let* $\{\lambda_i^A, \mathbf{v}_i^A\}_{i=1}^n$ *and* $\{\lambda_j^H, \mathbf{v}_j^H\}_{j=1}^d$ *be the eigenvalue and eigenvectors of* $\mathbf{A}$ *and* $\mathbf{H}$. *Let the eigenvalues (and associated eigenvectors) be sorted as follows,* $\lambda_1^A \leq \cdots \leq \lambda_n^A$ *and* $|1 + \lambda_1^H| \leq \cdots \leq |1 + \lambda_d^H|$. *Let* $\varphi_1^H, \ldots, \varphi_d^H$ *be the phases of* $\lambda_1^H, \ldots, \lambda_d^H$. *As the number of layers* $L \to \infty$, *one eigenvalue dominates the rest (multiple dominate if there are ties):*

$$
\begin{cases}
\left.\begin{matrix}
(1 + \lambda_d^H\lambda_n^A) & \text{if } |1 + \lambda_d^H\lambda_n^A| \geq 1 \\
(1 + \lambda_{\min}^H\lambda_1^A) & \text{if } |1 + \lambda_d^H\lambda_n^A| < 1
\end{matrix}\right\} & \text{if } \lambda_1^A > 0 \\[2em]
\left.\begin{matrix}
(1 + \lambda_d^H\lambda_n^A) & \text{if } |1 + \lambda_d^H\lambda_n^A| > |1 + \lambda_k^H\lambda_1^A| \\
(1 + \lambda_k^H\lambda_1^A) & \text{if } |1 + \lambda_d^H\lambda_n^A| < |1 + \lambda_k^H\lambda_1^A|
\end{matrix}\right\} & \text{if } \lambda_1^A < 0, \varphi_d^H \in [-\tfrac{\pi}{2}, \tfrac{\pi}{2}] \\[2em]
\left.\begin{matrix}
(1 + \lambda_d^H\lambda_n^A) & \text{if } |1 + \lambda_d^H\lambda_n^A| > |1 + \lambda_d^H\lambda_1^A| \\
(1 + \lambda_d^H\lambda_1^A) & \text{if } |1 + \lambda_d^H\lambda_n^A| < |1 + \lambda_d^H\lambda_1^A|
\end{matrix}\right\} & \text{if } \lambda_1^A < 0, \varphi_d^H \in (\tfrac{\pi}{2}, \pi] \cup [-\pi, -\tfrac{\pi}{2})
\end{cases}
$$

*where* $\lambda_{\min}^H$ *is the eigenvalue of* $\mathbf{H}$ *with smallest magnitude and* $\lambda_k^H$ *is the eigenvalue with the largest index* $k$ *such that* $\varphi_k^H \in (\pi/2, \pi] \cup [-\pi, -\pi/2)$.

*Proof.* Given Lemma 1, the eigenvalues and eigenvectors of $(\mathbf{I}+\mathbf{H}\otimes\mathbf{A})$ are equal to $(1+\lambda_j^H\lambda_i^A)$ and $\mathbf{v}_j^H\otimes\mathbf{v}_i^A$ for all $j\in\{1,...,d\}$ and $i\in\{1,\ldots,n\}$. Recall that eigenvalues (and associated eigenvectors) are sorted in the following order $\lambda_1^A\leq\cdots\leq\lambda_n^A$ and $|1+\lambda_1^H|\leq\cdots\leq|1+\lambda_d^H|$. Our goal is to understand the identity of the dominating eigenvalue(s) $\lambda_{j*}^H\lambda_{i*}^A$ for all possible values of $\lambda_H,\lambda_A$.

First recall that $\lambda_i^A\in(-1,1]$ and $\lambda_n^A=1$. A useful way to view selecting $\lambda_j^H\lambda_i^A$ to maximize $|1+\lambda_j^H\lambda_i^A|$ is as maximizing distance to $-1$. If (i), $\lambda_1^A>0$ then $\lambda_i^A$, for all $i\in\{1,\ldots,n-1\}$ always shrinks $\lambda_j^H$ to the origin and $\lambda_n^A$ leaves it unchanged. Because of how the eigenvalues are ordered we must have that $|1+\lambda_j^H|=|1+\lambda_j^H\lambda_n^A|\leq|1+\lambda_d^H\lambda_n^A|=|1+\lambda_d^H|$. If $|1+\lambda_d^H\lambda_n^A|\geq1$ then shrinking any $\lambda_i^H$ to the origin will also move it closer to $-1$. However, if $|1+\lambda_d^H\lambda_n^A|<1$ then shrinking to the origin can move $\lambda_i^H$ farther from $-1$ than $|1+\lambda_d^H\lambda_n^A|$. The eigenvalue of $\mathbf{H}$ that can be moved farthest is the one with the smallest overall magnitude, defined as $\lambda_{\min}^H$. The eigenvalue of $\mathbf{A}$ that can shrink it the most is $\lambda_1^A$. This completes the first two cases.

If instead (ii), $\lambda_1^A<0$ then it is possible to 'flip' $\lambda_j^H$ across the origin, and so the maximizer depends on $\varphi_j^H$. If (a) $\varphi_d^H\in[-\pi/2,\pi/2]$ then let $\lambda_k^H$ be the eigenvalue with the largest index $k$ such that $\varphi_k^H\in(\pi/2,\pi]\cup[-\pi,-\pi/2)$. It is possible that 'flipping' this eigenvalue across the origin makes it farther away than $\lambda_d^H$, i.e., $|1+\lambda_k^H\lambda_1^A|>|1+\lambda_d^H\lambda_n^A|$. In this case $(1+\lambda_k^H\lambda_1^A)$ dominates, otherwise $(1+\lambda_d^H\lambda_n^A)$ dominates. If they are equal then both dominate. If instead (b) $\varphi_d^H\in(\pi/2,\pi]\cup[-\pi,-\pi/2)$ then either $|1+\lambda_d^H\lambda_n^A|>|1+\lambda_{j'}^H\lambda_{i'}^A|$ for all $j'\neq d$ and $i'\neq n$, and so $(1+\lambda_d^H\lambda_n^A)$ dominates, or 'flipping' $\lambda_d^H$ increases its distance from $-1$, and so $|1+\lambda_d^H\lambda_1^A|>|1+\lambda_{j'}^H\lambda_{i'}^A|$ for all $j'\neq d$ and $i'\neq n$, and instead $(1+\lambda_d^H\lambda_1^A)$ dominates. Because we cannot have that $|1+\lambda_d^H\lambda_n^A|=|1+\lambda_d^H\lambda_1^A|$ as $\lambda_1^A>-1$ this covers all cases. $\qquad\square$

Now we can prove Theorem 1.

**Theorem 1.** *Given the Transformer update in eq. (3), let $\{\lambda_i^A\}_{i=1}^n$ and $\{\lambda_j^H\}_{j=1}^d$ be the eigenvalues of $\mathbf{A}$ and $\mathbf{H}$. Let the eigenvalues be sorted as follows, $\lambda_1^A\leq\cdots\leq\lambda_n^A$ and $|1+\lambda_1^H|\leq\cdots\leq|1+\lambda_d^H|$. As the number of layers $\ell\to\infty$, there are two types of dominating eigenvalues: (1) $(1+\lambda_{j*}^H\lambda_n^A)$. and (2) $(1+\lambda_{j*}^H\lambda_1^A)$*

The proof follows immediately from Lemma 2.

**Theorem 2.** *Given the Transformer update in eq. (3), if a single eigenvalue dominates, as the number of total layers $\ell\to\infty$, the feature representation $\mathbf{X}_\ell$ converges to one of two representations: (1) If $(1+\lambda_j^H\lambda_n^A)$ dominates then,*

$$\mathbf{X}_\ell\to(1+\lambda_j^H\lambda_n^A)^\ell s_{j,n}\mathbf{1}\mathbf{v}_j^{H^\top}, \tag{7}$$

*(2) If $(1+\lambda_j^H\lambda_1^A)$ dominates then,*

$$\mathbf{X}_\ell\to(1+\lambda_j^H\lambda_1^A)^\ell s_{j,1}\mathbf{v}_1^A\mathbf{v}_j^{H^\top} \tag{8}$$

*where $\mathbf{v}^H,\mathbf{v}^A$ are eigenvalues of $\mathbf{H},\mathbf{A}$ and $s_{j,i}:=\langle\mathbf{v}_{j,i}^{Q^{-1}},\mathsf{vec}(\mathbf{X})\rangle$ and $\mathbf{v}_{j,i}^{Q^{-1}}$ is row $ji$ in the matrix $\mathbf{Q}^{-1}$ (here $\mathbf{Q}$ is the matrix of eigenvectors of $(\mathbf{I}+\mathbf{H}\otimes\mathbf{A})$). (3) If multiple eigenvalues have the same dominating magnitude, $\mathbf{X}_\ell$ converges to the sum of the dominating terms.*

*Proof.* Recall that the eigenvalues and eigenvectors of $(\mathbf{I}+\mathbf{H}\otimes\mathbf{A})$ are equal to $(1+\lambda_j^H\lambda_i^A)$ and $\mathbf{v}_j^H\otimes\mathbf{v}_i^A$ for all $j\in\{1,...,d\}$ and $i\in\{1,\ldots,n\}$. This means,

$$\mathsf{vec}(\mathbf{X}_\ell)=\sum_{i,j}(1+\lambda_j^H\lambda_i^A)^\ell\langle\mathbf{v}_{j,i}^{Q^{-1}},\mathsf{vec}(\mathbf{X})\rangle(\mathbf{v}_j^H\otimes\mathbf{v}_i^A).$$

Recall that $\mathbf{v}_{j,i}^{Q^{-1}}$ is row $ji$ in the matrix $\mathbf{Q}^{-1}$, where $\mathbf{Q}$ is the matrix of eigenvectors $\mathbf{v}_j^H\otimes\mathbf{v}_i^A$. Further recall that $\mathbf{v}_i^A=\mathbf{1}$. As described in Theorem 1, as $\ell\to\infty$ at least one of the eigenvalues pairs $\lambda_j^H\lambda_i^A$ will dominate the expression $(1+\lambda_j^H\lambda_i^A)^\ell$, which causes $\mathsf{vec}(\mathbf{X}_L)$ to converge to the dominating term. Finally, we can rewrite, $\mathbf{v}_1\otimes\mathbf{v}_2$ as $\mathsf{vec}(\mathbf{v}_2\mathbf{v}_1^\top)$. Now all non-scalar terms have $\mathsf{vec}(\cdot)$ applied, so we can remove this function everywhere to give the matrix form given in eq. (7) and eq. (8). $\qquad\square$

**Corollary 1.** *If the residual connection is removed in the Transformer update, then the eigenvalues are of the form $(\lambda_j^H \lambda_i^A)$. Further, $(\lambda_{j*}^H \lambda_n^A)$ is always a dominating eigenvalue, and $\mathbf{X}_\ell \to$ $\mathbf{1}\left(\sum_{j* \in \mathcal{E}_{\max}^H} (\lambda_{j*}^H \lambda_n^A)^\ell s_{j*,n} \mathbf{v}_{j*}^{H\top}\right)$ as $\ell \to \infty$, where $\mathcal{E}_{\max}^H$ is the set of all eigenvalue indices equal to the dominating eigenvalue $\lambda_{j*}^H$.*

*Proof.* The eigendecomposition of the Transformer update without the residual connection is:

$$\mathsf{vec}(\mathbf{X}_\ell) = \sum_{i,j} (\lambda_j^H \lambda_i^A)^\ell \langle \mathbf{v}^{Q^{-1}}_{j,i}, \mathsf{vec}(\mathbf{X}) \rangle (\mathbf{v}_j^H \otimes \mathbf{v}_i^A).$$

In this case, $(\lambda_{j*}^H \lambda_n^A)$ is always a dominating eigenvalue because $|\lambda_n^A| > |\lambda_i^A|$ for any $i \in \{1, \ldots, n-1\}$. This observation, combined with the above eigendecomposition, produces $\mathbf{X}_\ell \to \mathbf{1}\left(\sum_{j* \in \mathcal{E}_{\max}^H} (\lambda_{j*}^H \lambda_n^A)^\ell s_{j*,n} \mathbf{v}_{j*}^{H\top}\right)$ as $\ell \to \infty$. $\qquad\square$

**Theorem 3.** *Given the Transformer update eq. (3), as the number of total layers $\ell \to \infty$, if **(1)** one eigenvalue $(1 + \lambda_j^H \lambda_n^A)$ dominates, we have input convergence, angle convergence, and rank collapse. If **(2)** one eigenvalue $(1 + \lambda_j^H \lambda_1^A)$ dominates, we do not have input convergence or angle convergence, but we do have rank collapse. If **(3)** multiple eigenvalues have the same dominating magnitude and: (a) there is at least one dominating eigenvalue $(1 + \lambda_{j*}^H \lambda_{i*}^A)$ where $\lambda_{i*}^A \neq \lambda_n^A$, then we do not have input convergence or angle convergence, or (b) the geometric multiplicity of $\lambda_1^A$ and $\lambda_{j*}^H$ are both greater than 1, then we also do not have rank collapse.*

*Proof.* If **(1)** one eigenvalue $(1 + \lambda_j^H \lambda_n^A)$ dominates then we have that $\mathbf{X}_\ell \to (1 + \lambda_j^H \lambda_n^A)^\ell s_{j,n} \mathbf{1} \mathbf{v}_j^{H\top}$. Therefore, $\mathbf{X}_\ell$ has all the same inputs which implies input convergence, angle convergence, and rank collapse. If **(2)** one eigenvalue $(1 + \lambda_j^H \lambda_1^A)$ dominates then we have that $\mathbf{X}_\ell \to (1 + \lambda_j^H \lambda_1^A)^\ell s_{j,1} \mathbf{v}_1^A \mathbf{v}_j^{H\top}$. Therefore, we do not have input convergence. Further as $\mathbf{v}_1^A$ can contain both positive an negative components we do not have angle convergence. However, $\mathbf{X}_\ell$ is rank one so we do have rank collapse. If **(3)** multiple eigenvalues have the same dominating magnitude and: (a) there is at least one dominating eigenvalue $(1 + \lambda_{j*}^H \lambda_{i*}^A)$ where $\lambda_{i*}^A \neq \lambda_n^A$ then we do not have input convergence or angle convergence, as shown for case (2); if (b) the geometric multiplicity of $\lambda_1^A$ and $\lambda_{j*}^H$ are both greater than 1, then $\mathbf{X}_\ell$ converges to the sum of at least 2 rank-1 matrices which are not themselves linear combinations of each other. Therefore, $\mathsf{rank}(\mathbf{X}_\ell) \geq 2$. $\qquad\square$

**Corollary 2.** *If the residual connection is removed in the Transformer update, input convergence, angle convergence, and rank collapse are guaranteed.*

*Proof.* Corollary 1 tells us that in this case $\mathbf{X}_\ell \to \mathbf{1}\left(\sum_{j* \in \mathcal{E}_{\max}^H} (\lambda_{j*}^H \lambda_n^A)^\ell s_{j*,n} \mathbf{v}_{j*}^{H\top}\right)$ as $\ell \to \infty$. This matrix is rank-1 and so we have input convergence, angle convergence, and rank collapse. $\qquad\square$

**Corollary 3.** *If the eigenvalues of $\mathbf{H}$ fall within $[-1, 0)$, then $(1 + \lambda_{j*}^H \lambda_1^A)$ dominates. If the eigenvalues of $\mathbf{H}$ fall within $(0, \infty)$, then $(1 + \lambda_{j*}^H \lambda_n^A)$ dominates.*

*Proof.* Let $\lambda_1^H \leq \cdots \leq \lambda_d^H$. Again we can think of selecting $\lambda_j^H \lambda_i^A$ that maximizes $|1 + \lambda_j^H \lambda_i^A|$ as maximizing the distance of $\lambda_j^H \lambda_i^A$ to $-1$. Consider the first case where $\lambda_1^H, \cdots, \lambda_d^H \in [-1, 0)$, and so $\lambda_1^H$ is the closest eigenvalue to $-1$ and $\lambda_d^H$ is the farthest. If $\lambda_1^A > 0$ then all $\lambda^A$ can do is shrink $\lambda^H$ to the origin, where $\lambda_1^A$ shrinks $\lambda^H$ the most. The closest eigenvalue to the origin is $\lambda_d^H$, and so $(1 + \lambda_d^H \lambda_1^A)$ dominates. If instead $\lambda_1^A < 0$, then we can 'flip' $\lambda_j^H$ over the origin, making it farther from $-1$ than all other $\lambda_{j'}^H$. The eigenvalue that we can 'flip' the farthest from $-1$ is $\lambda_1^H$, and so $(1 + \lambda_1^H \lambda_1^A)$ dominates. If all eigenvalues of $\mathbf{H}$ are equal, then both $(1 + \lambda_d^H \lambda_1^A)$ and $(1 + \lambda_1^H \lambda_1^A)$ dominate. For the second case where $\lambda_1^H, \cdots, \lambda_d^H \in (0, \infty)$, we have that $|1 + \lambda_d^H \lambda_n^A| > |1 + \lambda_{j'}^H \lambda_i^A|$ for all $j' \in \{1, \ldots, d-1\}$ and $i' \in \{1, \ldots, n-1\}$. This is because, by definition $\lambda_d^H \lambda_n^A > \lambda_{j'}^H \lambda_{i'}^A$. Further, $1 + \lambda_d^H \lambda_n^A \geq |1 + \lambda_{j'}^H \lambda_{i'}^A|$ as the largest $|1 + \lambda_{j'}^H \lambda_{i'}^A|$ can be is either (i) $|1 - \epsilon \lambda_d^H|$ for $0 < \epsilon < 1$ or (ii) $|1 + \lambda_{d-1}^H \lambda_n^A|$ (i.e., in (i) $\lambda_d^H$ is negated by $\lambda_1^A$ and in (ii) $\lambda_{d-1}^H$ is the next largest value of $\lambda^H$). For (i), it must be that $1 + \lambda_d^H \lambda_n^A \geq |1 - \epsilon \lambda_d^H|$ as $\lambda_d^H > 0$. For (ii) $\lambda_d^H \geq \lambda_{d-1}^H > 0$, and so $|1 + \lambda_d^H \lambda_n^A| \geq |1 + \lambda_{d-1}^H \lambda_n^A|$. Therefore $\lambda_n^A$ dominates. $\qquad\square$

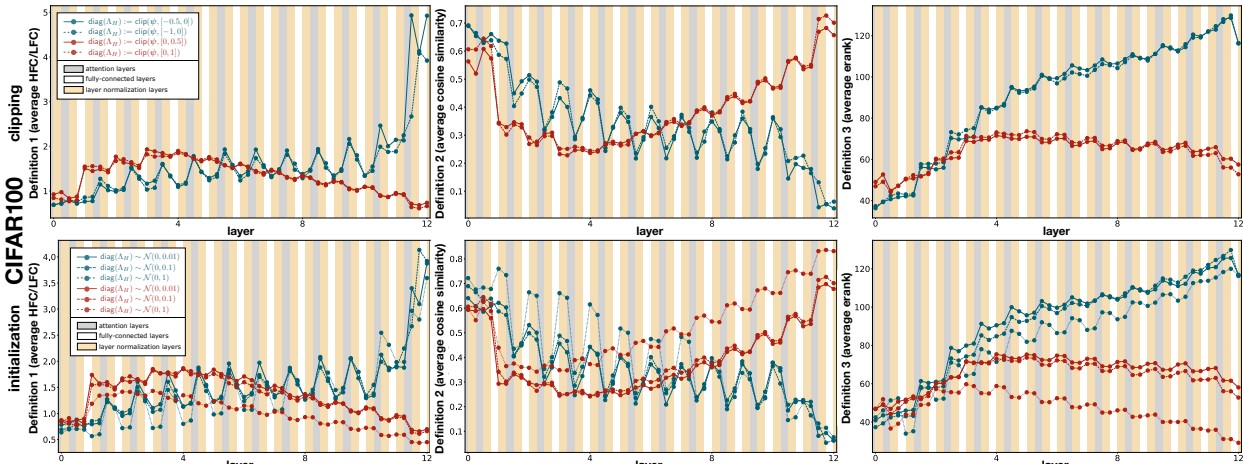

Figure 8: **Sensitivity of smoothing metrics.** HFC/LFC (Definition 1), cosine similarity (Definition 2), and effective rank (Definition 3) for reparameterized models with different clipping thresholds and initialization variances for $\mathsf{diag}(\Lambda_H)$.

## B  Training & Architecture Details

Crucially, even though our theoretical analysis applies for fixed attention **A** and weights **H**, **we use existing model architectures throughout**, i.e., including different attention/weights each layer, multi-head attention, layer normalization (arranged in the pre-LN format Xiong et al. (2020)), and fully-connected layers.[4]

**Image Classification: Training & Architecture Details.**   We base our image classification experiments on the ViT model Dosovitskiy et al. (2021) and training recipe introduced in Touvron et al. (2021a). On CIFAR100 for 300 epochs using the cross-entropy loss and the AdamW optimizer Loshchilov & Hutter (2019). Our setup is the one used in Park & Kim (2022) which itself follows the DeiT training recipe Touvron et al. (2021a). We use a cosine annealing schedule with an initial learning rate of $1.25 \times 10^{-4}$ and weight decay of $5 \times 10^{-2}$. We use a batch size of 96. We use data augmentation including RandAugment Cubuk et al. (2019), CutMix Yun et al. (2019), Mixup Zhang et al. (2018), and label smoothing Touvron et al. (2021a). The models were trained on two Nvidia RTX 2080 Ti GPUs. On ImageNet, we use the original DeiT code and training recipe described above. Changes from CIFAR100 are that we use a batch size of 512 and train on a single Nvidia RTX 4090 GPU.

**Text Generation: Training & Architecture Details.**   We base our NLP experiments on Geiping & Goldstein (2023), using their code-base. Following this work we pre-train encoder-only 'Crammed' Bert models with a maximum budget of 24 hours. We use a masked language modeling objective and train on the Pile dataset Gao et al. (2020). The batch size is 8192 and the sequence length is 128. We evaluate models on SuperGLUE Wang et al. (2020) after fine-tuning for each task. In order to ensure a fair comparison, all models are trained on a reference system with an RTX 4090 GPU. We use mixed precision training with bfloat16 as we found it to be the most stable Kaddour et al. (2023b).

## C  Hyperparameter Sensitivity

Another aspect that may have an effect on the smoothing behaviour of reparameterized models is the choice of hyperparameters. Specifically, there are two hyperparameters we set for $\mathsf{diag}(\Lambda_H)$: (a) the clipping interval (defaults: $\mathsf{clip}(\psi, [-1, 0])$ and $\mathsf{clip}(\psi, [0, 1])$), and (b) the initialization variance (default: 0.1). To test (a), we halve the size of each clipping interval. For (b), we try initialization variances within the set $\{0.01, 0.1, 1\}$.

---

[4]If a model has multiple heads we will define $\mathbf{W}_V = \mathbf{V}_H$ and $\mathbf{W}_{\mathsf{proj}} = \Lambda_H \mathbf{V}_H^\top$).

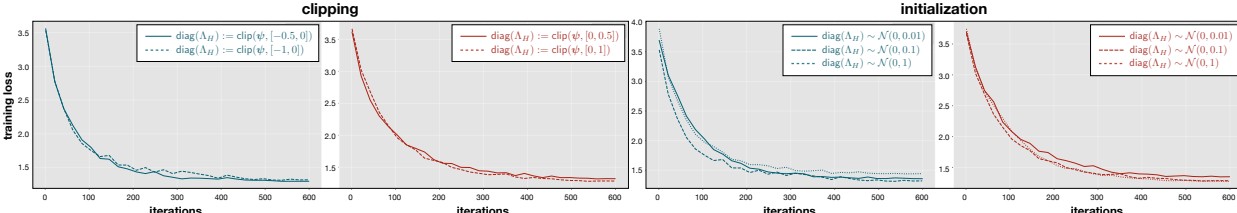

Figure 9: **Sensitivity of training loss.** Training loss curves for reparameterized models with different clipping thresholds and initialization variances for $\mathsf{diag}(\Lambda_H)$.

We test all changes on CIFAR100, as this dataset seems most sensitive to reparameterization and so should give us a sense of an upper bound on hyperparameter sensitivity. The results are shown in Figure 8. We notice that clipping interval seems to have a small effect on the smoothing behaviour of reparameterized models: the largest change is the final HFC/LFC of the sharpening model, which increases by roughly one when the clipping interval is halved.

The initialization variance has a much larger effect on the smoothing model: increasing the variance induces stronger smoothing across all metrics, for all layers beyond the second layer. Curiously it has little effect on the final smoothing of the sharpening model, even when increasing the variance reduces sharpening for intermediate layers. Finally, we were curious if hyperparameter choices introduced any training instability which could explain these results. Figure 9 shows that this is not the case: under any hyperparameter choice tested, the training curves are stable.

## D  Distribution of the eigenvalues of $\mathbf{H}$ in trained models

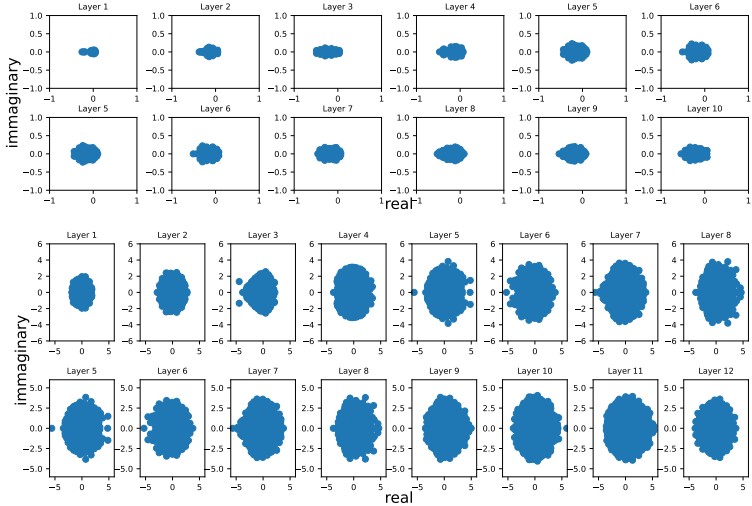

Figure 10: **Distributions of eigenvalues of H** (*Top*) Vision models have distributions skewing to the negatives; (*Bottom*) Language models have symmetrically distributed eigenvalues.

