# OpenReview forum: "Setting the Record Straight on Transformer Oversmoothing"
_TMLR — Accepted by TMLR_

### Review · Reviewer_eQhT · 2025-01-11

**Summary Of Contributions:**

This paper provides both theoretical analyses and empirical experiments on the oversmoothing phenomenon in Transformer architectures. The authors summarize three metrics from related work to measure oversmoothing: input convergence, angle convergence, and rank collapse. They investigate smoothing behavior by analyzing the eigenspectrum of attention and weight matrices. However, the study excludes layer normalization and feedforward layers, as these components complicate the analysis. Instead, the authors propose a reparameterization of the weight matrices and explore how it influences smoothing behavior. While the results could be more clearly presented, the authors appear to argue that oversmoothing is not inevitable and can be controlled through this reparameterization.

**Audience:**

Yes

**Broader Impact Concerns:**

No broader impact concern.

**Claims And Evidence:**

Yes

**Requested Changes:**

Critical to securing my recommendation for acceptance:
- Ensure proper usage of reference citation styles, distinguishing between `(Author, Year)` and `Author (Year)`.
- The fonts in Figures 2 and 3 are too small, making them difficult to read. Additionally, these figures need more detailed explanations to clarify how the conclusions are drawn.

Simply strengthen the work in my view:
- Improve the overall organization and writing of the paper to enhance clarity and readability.
- The purpose of Section 5, which introduces the reparameterization, is unclear. It seems the conclusions vary across different datasets, but this needs to be explained more explicitly.
- Clearly distinguish between the novel contributions of this paper and the ideas summarized from existing work to better highlight the originality of the study.

**Strengths And Weaknesses:**

Strengths:
- The topic of Transformer oversmoothing is important and merits further study.
- The paper provides a comprehensive summary of related work, particularly on the methods used to measure oversmoothing.
- The use of eigenspectrum analysis and the proposed reparameterization approach seem to be novel.

Weaknesses:
- The paper's organization and clarity could be improved. While I am not an expert in this field, certain sections are difficult to follow, which may hinder broader accessibility.
- This paper excludes layer normalization and feedforward layers, as these components complicate the analysis.
- The experimental results are not clearly explained.

---

> ### Author Response · Authors · 2025-04-24
> **Author Response**
>
> Thank you for your positive feedback, we respond to your requests below.
>
> > (Requested Changes - Critical 1) [.Ensure proper usage of reference citation styles, distinguishing between (Author, Year) and Author (Year)..]
>
> Thanks for catching this, we have fixed this.
>
>
> > (Requested Changes - Critical 2) [..The fonts in Figures 2 and 3 are too small, making them difficult to read. Additionally, these figures need more detailed explanations to clarify how the conclusions are drawn..]
>
> We have increased these font sizes. Further, we have added more detailed explanations to the captions. We hope this clears things up, please let us know if you would like further changes.
>
>
> > Weaknesses 1) [..The paper's organization and clarity could be improved. While I am not an expert in this field, certain sections are difficult to follow, which may hinder broader accessibility..]
>
> > (Requested Changes - Strengthen 1) [..Improve the overall organization and writing of the paper to enhance clarity and readability..]
>
> Thank you for this. We have completely restructured Section 5 to clarify the narrative. Additionally, we have added takeaway boxes to Sections 3 and 5, and renamed Section 4 to add clarification about the main results of these sections.
>
> > (Requested Changes - Strengthen 2) [..The purpose of Section 5, which introduces the reparameterization, is unclear. It seems the conclusions vary across different datasets, but this needs to be explained more explicitly..]
>
> > (Weaknesses 3) [..The experimental results are not clearly explained..]
>
> Thanks for this. Our aim in Section 5 was to show that our theory provides some explanatory power, through the reparameterizations, for how the Transformer block influences smoothing/sharpening in full Transformer models. We observed that the reparameterization did allow us to increase/reduce smoothing for models on CIFAR100 and The Pile. However, the influence is much more limited on the ImageNet model we tested. This then led us to investigate how layer normalization influences simplified models in Figure 4. As layer normalization weights seem to reduce or even flip smoothing we tested the full ImageNet model when layer normalization weights were removed and found that the reparameterizations, particularly the sharpening model, have a much larger effect, confirming our hypothesis that layer normalization has an influence on smoothing in full models (further evidence of this is shown in Tables 1-3). Finally, we were curious if the reparameterization had any effect on test error. We found that surprisingly it did improve performance for CIFAR100 but not for either the ImageNet model or Crammed BERT. Ultimately, this is okay: our goal was not to develop better performing models, but to try to understand whether smoothing is a fundamental problem for Transformers. Based on our findings, we believe it is not.
>
> Thank you for allowing us to clarify this. We agree that reformulating Section 5 to make this story clearer is a good idea. To do so, we have restructured Section 5 so that the layer normalization analysis appears directly after the reparameterization result and have added additional text to clarify the motivation of experiments. Thank you for your feedback.
>
> > (Requested Changes - Strengthen 3) [..Clearly distinguish between the novel contributions of this paper and the ideas summarized from existing work to better highlight the originality of the study..]
>
> We have added our contributions to the discussion section. Theoretically, we presented a new analysis detailing how the eigenspectrum of attention and weight matrices influences smoothing behavior across three different metrics. Prior work analyzed one metric in isolation. However, because our eigenvalue analysis uncovers the representations that oversmoothing models asymptotically converge to, we can analyze any of the primary three metrics. Empirically, we found that, contrary to prior findings, oversmoothing is not inevitable, even in existing pre-trained models. We have additionally empirically analyzed the impact of layer normalization and discovered the role of LN weights can play on smoothing behaviour.
>
> **response continues below**

---

> > ### Author Response · Authors · 2025-04-24
> > **Author Response (continued)**
> >
> > > (Weaknesses 3) [..This paper excludes layer normalization and feedforward layers, as these components complicate the analysis..]
> >
> > Thank you for this. To further investigate how tightly the analysis predicts full Transformer behaviour we test if we can better explain the impact of layer normalization. Specifically, Figure 4 seems to indicate that if the weights of layer normalization are negative then it can flip the intended behaviour of sharpening/smoothing parameterizations for Pre-LN models (all of the full Transformer models we test use Pre-LN). Further, positive weights can also reduce the smoothing behaviour of the smoothing parameterization. To better understand the impact of layer normalization, we remove the weights of layer normalization for our parameterizations on ImageNet in Figure 5 and Table 4. We find that for nearly all oversmoothing measures, the sharpening model resists oversmoothing even more strongly when layer normalization weights are removed (shown in the blue dotted lines in Figure 5). Curiously, the smoothing model with removed layer normalization weights has little impact on smoothing behaviour. These results provide some evidence that layer normalization impacts smoothing behaviour, and that this can be partially mitigated in full, finite-depth models by removing layer normalization weights.

---

### Review · Reviewer_1HFL · 2025-01-29

**Summary Of Contributions:**

Prior works observed that oversmoothing effect happens in transformer models when depth is increased, meaning that representations collapse: either to the same vector, to the same angle or to representations of rank 1. This gives unclear picture why transformers perform so well in practice despite this phenomenon. The main downside is a lot of assumptions and simplifications to perform this analysis in prior works which creates a huge gap between theory and practice. Authors e.g. show empirical analysis of oversmoothing in 3 different models / data and show inconsistency with theory for them.

The paper is trying to reduce this gap and perform more general analysis.
- First, authors still consider simplification of the transformer, but only removing the layer normalization and mlp (e.g., there is still residual connection + there are no assumptions on the attention matrix). With that authors show that the important piece for theory and analysis is the eigenspectrum of value and projection matrices, and depending on the spectrum and what eigenvalue dominates we will end with different oversmoothing regimes, but also may not end up e.g. with input or angle collapse, only with rank collapse, or even no collapse at all if there are multiple eigenvalues with the same dominating amplitude. These results are new and contradict prior work theory: new insights extend prior work findings for more general cases, reducing the gap between theory and practice.
- Second, authors discuss that layer norms and mlp could not give any theoretical results unless we make some assumptions. However we could probe the finding if spectrum is the cause of oversmoothing behaviour (or there is correlation) in practice by reparametrization of the value and projection matrices. I found this to be a smart and cool approach: authors inherit conditions of the spectrum which gives us sharpening or smoothing and now with reparametrization we can enforce either regime in the model to see what happens in practice with a real transformer. With that authors observe different behaviours and provide some analysis for the models. This is a way to get correlation between theoretical results (when simplifications are made) and empirical results.

Also, an interesting result for me was the corollary that without residuals we are always going to collapse, pointing out again why residuals are so important in practice.

**Audience:**

Yes

**Broader Impact Concerns:**

No any concerns, it is theoretical work on understanding how transformers work in general, which may impact the society later only.

**Claims And Evidence:**

No

**Requested Changes:**

The paper is really well written with good math notation and clear derivations for all theorems, including Appendix. I did not find any issues so far with derivations and claims in theory. I also like figure 1 a lot, this is a good summary. And the overall text is written as a well organized story, which will be simple to follow and understand for people outside theory or who want to get the insight w/o reading theoretical parts and proofs.

While I enjoyed the math and overall theoretical claims, I was expecting a bit more on the empirical side. Empirical results in Fig 3 did not convince me that we have a clear correlation between what is stated in theory and what is observed in practice. With that I believe, either more discussion of the results is needed, or maybe authors could reformulate a bit what they are trying to tel exactly by Sec 5. I think I quite did not get the outcomes as mainly all results for the baselines are quite opposite behaviours. I feel at the end of the paper I lost the thread of the results and consequences.

Minor typos / small suggestions to improve a bit the text:
- "while rank collapse is likely, it is also not required." -> "while rank collapse is likely, it is also not necessarily happening." (required is a bit weird to me from language reading).
- notation on 1 vector will be good to have, as I interpreted it first as a matrix with all ones, not the vector.
- figures 2, 3 - make the legend bigger, it is unreadable right now.
- eq (4) may be good to include "... vnA vjH  = ... 1 vjH (by using proposition 1)" - so explicitly point why there is "1" and not the "v" vector.
- initialization page 9: will your initialization give the final H to be the same initialization as for the baselines? does this restrict that you consider only particular initialization?

**Strengths And Weaknesses:**

**Strengths**
- theoretical results on getting the dependence of collapse based on the eigenspectrum of values and projection matrix with less assumptions as prior works
- new insights when oversmoothing happens, including cases when oversmoothing can be avoided in the models.
- attempt to investigate / prove transfer of theory to more general empirical settings via reparametrization and mechanism of enforcing oversmoothing or sharpening

**Weaknesses**
- Empirical results for the baseline models are not showing strong patterns of oversmoothing. I agree that different models and data give different trends, but all trends are not really drastic to say that models do or not do strong oversmoothing.
- Empirical results with controlling smoothness and sharpness show some changes but again it seems that it is not showing general correlation between eigenspectrum and empirical behaviour.

---

> ### Author Response · Authors · 2025-04-24
> **Author Response**
>
> Thank you for your interest in the paper and positive comments, we address your points below.
>
> > (Weakness 1) [..Empirical results for the baseline models are not showing strong patterns of oversmoothing. I agree that different models and data give different trends, but all trends are not really drastic to say that models do or not do strong oversmoothing..]
>
> We absolutely agree. This is actually the final straw that motivated us to investigate the theoretical analyses of past work that argued oversmoothing is inevitable. What we found is that different models and different datasets produce wildly different results, which contradicts the narrative of oversmoothing as an inherent issue in Transformer models.
>
> > (Weakness 2) [..Empirical results with controlling smoothness and sharpness show some changes but again it seems that it is not showing general correlation between eigenspectrum and empirical behaviour..]
>
> > (Requested Changes) [.. I was expecting a bit more on the empirical side. Empirical results in Fig 3 did not convince me that we have a clear correlation between what is stated in theory and what is observed in practice..]
>
> Thank you for this. We believe this is likely due to the impact of layer normalization. Specifically, Figure 4 seems to indicate that if the weights of layer normalization are negative then it can flip the intended behaviour of sharpening/smoothing parameterizations for Pre-LN models (all of the full Transformer models we test use Pre-LN). Further, positive weights can also reduce the smoothing behaviour of the smoothing parameterization. To better understand the impact of layer normalization, we remove the weights of layer normalization and plot the results of our parameterizations on ImageNet in Figure 5 and Table 4. We find that for nearly all oversmoothing measures, the sharpening model resists oversmoothing even more strongly when layer normalization weights are removed (shown in the blue dotted lines in Figure 5). Curiously, the smoothing model with removed layer normalization weights has little impact on smoothing behaviour. These results provide some evidence that layer normalization impacts smoothing behaviour, and that this can be partially mitigated in full, finite-depth models by removing layer normalization weights.
>
> > (Requested Changes) [..I believe, either more discussion of the results is needed, or maybe authors could reformulate a bit what they are trying to tel exactly by Sec 5. I think I quite did not get the outcomes as mainly all results for the baselines are quite opposite behaviours. I feel at the end of the paper I lost the thread of the results and consequences..]
>
> Thanks for this. Our aim in Section 5 was to show that our theory provides some explanatory power, through the reparameterizations, for how the Transformer block influences smoothing/sharpening in full Transformer models. We observed that the reparameterization did allow us to increase/reduce smoothing for models on CIFAR100 and The Pile. However, the influence is much more limited on the ImageNet model we tested. This then led us to investigate how layer normalization influences simplified models in Figure 4. As layer normalization weights seem to reduce or even flip smoothing we tested the full ImageNet model when layer normalization weights were removed and found that the reparameterizations, particularly the sharpening model, have a much larger effect, confirming our hypothesis that layer normalization has an influence on smoothing in full models (further evidence of this is shown in Tables 1-3). Finally, we were curious if the reparameterization had any effect on test error. We found that surprisingly it did improve performance for CIFAR100 but not for either the ImageNet model or Crammed BERT. Ultimately, this is okay: our goal was not to develop better performing models, but to try to understand whether smoothing is a fundamental problem for Transformers. Based on our findings, we believe it is not.
>
> Thank you for allowing us to clarify this. We agree that reformulating Section 5 to make this story clearer is a good idea. To do so, we have restructured Section 5 so that the layer normalization analysis appears directly after the reparameterization result and have added additional text to clarify the motivation of experiments. Thank you for your feedback.
>
> > (Requested Changes) [..Minor typos / small suggestions to improve a bit the text:..]
>
> Thank you for these suggestions and catching these typos!
>
> > [.."while rank collapse is likely, it is also not required." -> "while rank collapse is likely, it is also not necessarily happening." (required is a bit weird to me from language reading)..]
>
> Fixed.
>
> > [..notation on 1 vector will be good to have, as I interpreted it first as a matrix with all ones, not the vector..]
>
> Good call, we added a definition when this is first introduced in the second paragraph of Section 2.2.
>
> **response continues below**

---

> > ### Author Response · Authors · 2025-04-24
> > **Author Response (continued)**
> >
> > > [..figures 2, 3 - make the legend bigger, it is unreadable right now..]
> >
> > Agreed, fixed.
> >
> > > [..eq (4) may be good to include "... vnA vjH = ... 1 vjH (by using proposition 1)" - so explicitly point why there is "1" and not the "v" vector..]
> >
> > We have a slight stylistic preference for the original format as it aligns the expression above the sharpening expression which we think helps make visual comparison easier. That said, if you feel strongly about this we are happy to change it.
> >
> > > [..initialization page 9: will your initialization give the final H to be the same initialization as for the baselines? does this restrict that you consider only particular initialization?..]
> >
> > Good question. The initialization will liekly differ from the baselines. To judge the impact of our initialization strategy we plotted the three smoothing metrics when the initialization variance of $\mathsf{diag}(\Lambda_H)$ is varied within $(0.01,0.1,1)$ in Figure 6 (we also try varying the clipping threshold of $\mathsf{diag}(\Lambda_H)$). We find that the initialization variance can have a large effect on the smoothing model: increasing the variance induces stronger smoothing across all metrics, for all layers beyond the second layer. Curiously it has little effect on the final smoothing of the sharpening model, even when increasing the variance reduces sharpening for intermediate layers. We also track the impact of hyperparameter choices on the training loss in Figure 7. We find that any of the tested hyperparameter choices still lead to stable training curves. Thank you for asking about this, we have added this discussion to Section 5.

---

> > > ### Comment · Reviewer_1HFL · 2025-05-02
> > > **Reply**
> > >
> > > Dear Authors,
> > >
> > > Thanks for the comments, clarifications and additional analysis. Will go over revision in upcoming days. So far your responses looks good to me.
> > >
> > > Only one comment for now:
> > > > [..eq (4) may be good to include "... vnA vjH = ... 1 vjH (by using proposition 1)" - so explicitly point why there is "1" and not the "v" vector..]
> > >
> > > I got what you wanted. No strong preferences, more like readability w/o questioning why this equation looks like this. How about adding the footnote then? So that main text is aligned as you wanted, but then if derivation is at question footnote will make it obvious for transition?
> > >
> > > Thanks!

---

> > > > ### Comment · Reviewer_1HFL · 2025-05-02
> > > > **Question about revision details**
> > > >
> > > > By any chance do you have version where it is colored marked what changes you made? or at least list of sections / subsections changed?
> > > >
> > > > Thanks!

---

> > > > > ### Author Response · Authors · 2025-05-07
> > > > >
> > > > > Thank you for your response.
> > > > >
> > > > > > How about adding the footnote then?
> > > > >
> > > > > That's a great suggestion. We've added the footnote and believe it enhances the clarity of the derivation.
> > > > >
> > > > > > By any chance do you have version where it is colored marked what changes you made?
> > > > >
> > > > > We have updated the PDF to include a version with our changes highlighted in blue. Here is a summary of those changes:
> > > > > - added takeaway paragraphs to Sections 3 and 5 to improve the narrative flow of the paper
> > > > > - revised the title of Section 4 to more clearly indicate our objective of testing the theory
> > > > > - reformatted Tables 3, 4, and 5 for improved readability
> > > > > - rewrote most of section 5 for greater clarity
> > > > > - added Figures 5, 6, and 7, which illustrate the metrics for models with weightless Layer Normalization (LN), models with various initializations, and the corresponding loss curves for these differently initialized models.
> > > > >
> > > > > Thanks.

---

> > > > > > ### Comment · Reviewer_1HFL · 2025-05-11
> > > > > > **Reply**
> > > > > >
> > > > > > Dear Authors,
> > > > > >
> > > > > > Thanks a lot for the changes marked with the color -- simple to check! I did a pass and I am happy with the current version, now it reads very smooth and I think message in the empirical section is properly shown.
> > > > > >
> > > > > > Thanks for including results on the hyperparameters variations, I found it is helpful and interesting -- having models stable training but at the same time seeing again importance of initialization - I think this now gives one extra argument into understanding why initialization is so important and how it affects the model properties.
> > > > > >
> > > > > > **[Out of paper submission and revision]**
> > > > > >
> > > > > > One question which came up during proof reading - when you remove LN what do you think still affects model that smoothing is not changed much? Also do you think that your observation on the effect of pre-LN vs post-LN where pre-LN doesn't change smoothing e.g. can explain partially why pre-LN is more stable arch for deep models than post-LN? Can we actually make connection between expressiveness of the model and smoothing / sharpening properties in the end as we know post-LN generalizes better but harder to train (maybe we want to have model be able to change / jump between smooth / sharp states or at least having it simply to do so)?

---

> ### Author Response · Authors · 2025-05-19
> **Author Response**
>
> > [..Thanks a lot for the changes marked with the color -- simple to check! I did a pass and I am happy with the current version, now it reads very smooth and I think message in the empirical section is properly shown..]
> > [..Thanks for including results on the hyperparameters variations, I found it is helpful and interesting -- having models stable training but at the same time seeing again importance of initialization - I think this now gives one extra argument into understanding why initialization is so important and how it affects the model properties..]
>
> You’re welcome! Thank you for the insightful suggestions.
>
> > [..[Out of paper submission and revision]..]
>
> Thank you for the thoughtful follow-up.
>
> > [..when you remove LN what do you think still affects model that smoothing is not changed much?..]
>
> This is an interesting question. There are a few possible explanations:
> 1. Lingering effects of LN: As we did not fully remove LN, but just fixed the scale and shift parameters to $\gamma = 1$ and $\beta = 0$ (we have updated the text in Section 5 to make this clearer), it is possible that the normalization of LN still has a lingering effect in full models that does not show up in the repeated models of Figure 4.
> 2. Initialization: The default initialization of diag$(\Lambda_H)$ of $\mathcal{N}(0,0.1)$ may be too small and may reduce how much smoothing can occur. If the initialization is instead set to $\mathcal{N}(0,1)$, great smoothing occurs for CIFAR100, as shown in Figure 6.
> 3. Unmodelled effects of feed-forward layers: It's possible feed-forward layers have asymmetric impacts on sharpening and smoothing behaviour.
>
> > [..Also do you think that your observation on the effect of pre-LN vs post-LN where pre-LN doesn't change smoothing e.g. can explain partially why pre-LN is more stable arch for deep models than post-LN?..]
>
> For this we would need to test the above explanations and if none (and possibly others) seemed to have an effect then we would need to test whether there was a relationship between stability and smoothing behaviour. At the moment we don't have enough information to confidently speculate one way or the other.
>
> > [..Can we actually make connection between expressiveness of the model and smoothing / sharpening properties in the end as we know post-LN generalizes better but harder to train (maybe we want to have model be able to change / jump between smooth / sharp states or at least having it simply to do so)?..]
>
> Another curious point! We have looked at whether a model leans towards smoothing or sharpening but haven't looked into whether a model can adapt its smoothing behaviour at test time. What our theory shows is that since the attention matrix varies, as long as the channel mixing matrix does not have fully positive or fully negative eigenvalues, it is possible for the model to be smoothing or sharpening depending on the spectrum of the attention matrix. The result is that, while strictly speaking the models can "jump" between smoothing and sharpening, they can only do so currently in a way that significantly affects the attention patterns. It is possible to come up with attention variants that are able to do this in a less restrictive way. One could do smoothing and sharpening simultaneously like the differential transformer [1], which can be thought of as learning one smoothing and one sharpening attention matrix, or sequentially like blurring sharpening models, of which GFSA [2] is a generalization.
>
> [1] Ye et al. Differential Transformer. ICLR, 2025.
>
> [2] Choi et al. 2023. Graph Convolutions Enrich the Self-Attention in Transformers! NeurIPS, 2023.

---

> > ### Comment · Reviewer_1HFL · 2025-05-21
> > **Reply**
> >
> > Thanks for detailed thoughts! All makes sense to me and yep, agree on extra experiments are needed. Looking forward to see future work (if any) on these points! Maybe we can get better understanding about the generalization and training stability in the end :).
> >
> > Thanks again for the thoughtful discussion over the review process!
> >
> > Best,
> >
> > Reviewer.

---

> > > ### Author Response · Authors · 2025-06-10
> > > **Thank you**
> > >
> > > Thank you again for your detailed feedback Reviewer 1HFL. It is rare to see this and we appreciate what we learned from it!
> > >
> > > We just realized that the updated submission extended beyond the regular submission length of 12 pages (to 13), so we have moved the hyperparameter sensitivity study to the appendix. We hope this is alright!
> > >
> > > Thank you again!
> > >
> > > - The Authors

---

### Review · Reviewer_MUFK · 2025-04-17

**Summary Of Contributions:**

This paper revisits the phenomenon of oversmoothing in deep Transformer models, challenging the prevailing view that increasing depth inevitably causes features to converge (in input space), angles to collapse, or rank to drop to one. The authors:
1. Empirically show that several pre‑trained vision and language Transformers do not monotonically oversmooth according to the three common metrics (input convergence, angle convergence, rank collapse).
2. Develop a theoretical analysis of a simplified Transformer update (fixed attention, single head, residual connection) via eigenspectrum arguments. They prove that whether a model oversmooths depends on the dominant eigenvalue of the Kronecker‑sum matrix, yielding precise conditions under which each of the three collapse phenomena do—or do not—occur.
3. Derive a simple reparameterization of the residual‐weight matrix (via a clipped diagonal eigenvalue spectrum) that provably biases the model toward sharpening or smoothing behavior.
4. Validate this reparameterization on vision (ViT variants) and language (“crammed” BERT) architectures, showing that one can indeed shift smoothing metrics—and sometimes improve downstream accuracy in small‐scale settings.

**Audience:**

Yes

**Claims And Evidence:**

Yes

**Requested Changes:**

**Requested Changes:**

The paper would benefit from systematic ablations to isolate the effects of its key hyperparameters. For instance, varying the initial scale of the diagonal entries of $\(\Lambda_H\)$ could reveal whether the observed smoothing and sharpening trends are robust to the initialization variance. Likewise, experimenting with different clipping intervals (e.g.$[-0.5,0]$ versus $[-1,0]$ may uncover the sensitivity of model behavior to the allowed eigenvalue range. Finally, tracking the impact of these choices on the training‐loss convergence rate would clarify whether constraining the spectrum hinders or accelerates optimization.

**Strengths And Weaknesses:**

**Strengths**
1. The eigenspectrum analysis is novel and unifies prior (sometimes conflicting) results. It identifies exactly when input convergence, angle collapse, and rank collapse are necessary versus merely possible.
2. By directly constraining the eigenvalues of the residual‐weight matrix, practitioners gain a simple knob for controlling smoothing behavior. The empirical results confirm that this knob indeed moves the smoothing metrics.

**Weaknesses**
1. The theoretical model omits layer normalization, feedforward layers, multi‑head attention, positional encodings, and (for language models) causal masking. While the authors acknowledge these gaps, it remains unclear how tightly the simplified analysis predicts behavior in actual Transformer blocks.
2. The asymptotic nature of the results ($\ell \to \infty$) may not map cleanly onto finite‐depth networks (e.g., 12–48 layers).
3. Constraining eigenvalues via clipping may introduce gradient instability or hamper learning dynamics; the paper does not thoroughly analyze training stability (beyond noting that large initial $\(\Lambda_H\)$ can blow up).

---

> ### Author Response · Authors · 2025-04-24
> **Author Response**
>
> Thank you for your thoughtful comments, we hope to address your concerns below.
>
> > (Weakness 1) [..The theoretical model omits layer normalization, feedforward layers, multi‑head attention, positional encodings, and (for language models) causal masking. While the authors acknowledge these gaps, it remains unclear how tightly the simplified analysis predicts behavior in actual Transformer blocks..]
>
> > (Weakness 2) [..The asymptotic nature of the results (can blow up).) may not map cleanly onto finite‐depth networks (e.g., 12–48 layers)..]
>
> Thank you for this. To further investigate how tightly the analysis predicts full Transformer behaviour we test if we can better explain the impact of layer normalization. Specifically, Figure 4 seems to indicate that if the weights of layer normalization are negative then it can flip the intended behaviour of sharpening/smoothing parameterizations for Pre-LN models (all of the full Transformer models we test use Pre-LN). Further, positive weights can also reduce the smoothing behaviour of the smoothing parameterization. To better understand the impact of layer normalization, we remove the weights of layer normalization for our parameterizations on ImageNet in Figure 5 and Table 4. We find that for nearly all oversmoothing measures, the sharpening model resists oversmoothing even more strongly when layer normalization weights are removed (shown in the blue dotted lines in Figure 5). Curiously, the smoothing model with removed layer normalization weights has little impact on smoothing behaviour. These results provide some evidence that layer normalization impacts smoothing behaviour, and that this can be partially mitigated in full, finite-depth models by removing layer normalization weights.
>
> > (Weakness 3) [..Constraining eigenvalues via clipping may introduce gradient instability or hamper learning dynamics; the paper does not thoroughly analyze training stability (beyond noting that large initial can blow up)..]
>
> > (Requested Changes) [..The paper would benefit from systematic ablations..For instance, varying the initial scale of the diagonal entries of $(\Lambda_H)$..Likewise, experimenting with different clipping intervals.. Finally, tracking the impact of these choices on the training‐loss convergence rate would clarify whether constraining the spectrum hinders or accelerates optimization..]
>
> This is a good question. We have added additional results investigating these things in Figures 6 and 7. We notice that the clipping interval seems to have a small effect on the smoothing behaviour of reparameterized models: the largest change is the final HFC/LFC of the sharpening model, which increases by roughly one when the clipping interval is halved. The initialization variance has a much larger effect on the smoothing model: increasing the variance induces stronger smoothing across all metrics, for all layers beyond the second layer. Curiously it has little effect on the final smoothing of the sharpening model, even when increasing the variance reduces sharpening for intermediate layers. We also track the impact of hyperparameter choices on the training loss. We find that any of the tested hyperparameter choices still lead to stable training curves. Thank you for this suggestion, we have added this discussion to Section 5.

---

### Decision · Action_Editor_RG9a · 2025-06-11

**Recommendation:** Accept as is

**Audience:**

Yes

**Audience Explanation:**

This work provides both theoretical and empirical analysis of the Transformer model. Transformer is now used by mainstream domains such as computer vision and language, which covers a significant range of applications. Given the wide usage of the Transformer, I believe a large number of individuals in TMLR's audience will be interested in knowing the findings of this paper.

**Claims And Evidence:**

Yes

**Claims Explanation:**

This work provides both theoretical and empirical analysis of the Transformer model. The theoretical claims are accurate, with clearly stated assumptions and backed up by experiments.